

# Can Large Language Model Agents Simulate Human Trust Behavior?

**Chengxing Xie**[*1, 11]  **Canyu Chen**[*2]
**Feiran Jia**[4]  **Ziyu Ye**[5]  **Shiyang Lai**[5]  **Kai Shu**[6]  **Jindong Gu**[3]  **Adel Bibi**[3]  **Ziniu Hu**[7]
**David Jurgens**[8]  **James Evans**[5, 9, 10]  **Philip H.S. Torr**[3]  **Bernard Ghanem**[1]  **Guohao Li**[†3, 11]

[1]KAUST  [2]Illinois Institute of Technology  [3]University of Oxford  [4]Pennsylvania State University
[5]University of Chicago  [6]Emory  [7]California Institute of Technology
[8]University of Michigan  [9]Santa Fe Institute  [10]Google  [11]CAMEL-AI.org

Project website: https://agent-trust.camel-ai.org

## Abstract

Large Language Model (LLM) agents have been increasingly adopted as simulation tools to model humans in social science and role-playing applications. However, one fundamental question remains: *can LLM agents really simulate human behavior?* In this paper, we focus on one critical and elemental behavior in human interactions, *trust*, and investigate whether LLM agents can simulate human trust behavior. We first find that LLM agents generally exhibit trust behavior, referred to as ***agent trust***, under the framework of *Trust Games*, which are widely recognized in behavioral economics. Then, we discover that GPT-4 agents manifest high ***behavioral alignment*** with humans in terms of trust behavior, indicating *the feasibility of simulating human trust behavior with LLM agents*. In addition, we probe the biases of agent trust and differences in agent trust towards other LLM agents and humans. We also explore the intrinsic properties of agent trust under conditions including external manipulations and advanced reasoning strategies. Our study provides new insights into the behaviors of LLM agents and the fundamental analogy between LLMs and humans beyond *value alignment*. We further illustrate broader implications of our discoveries for applications where trust is paramount.

## 1 Introduction

There is an increasing trend to adopt Large Language Models (LLMs) as agent-based simulation tools for humans in various social science fields including economics, politics, psychology, ecology and sociology (Gao et al., 2023b; Manning et al., 2024; Ziems et al., 2023), and role-playing applications such as assistants, companions and mentors (Yang et al., 2024; Abdelghani et al., 2023; Chen et al., 2024) due to their human-like cognitive capacity. Nevertheless, most previous research is based on one insufficiently validated assumption that LLM agents behave like humans in simulation. Thus, a fundamental question remains: *Can LLM agents really simulate human behavior?*

In this paper, we focus on *trust* behavior in human interactions, which comprises the intention to place self-interest at risk based on the positive expectations of others (Rousseau et al., 1998). Trust is one of the most critical and elemental behaviors in human interactions and plays an essential role in social settings ranging from daily communication to economic and political institutions (Uslaner, 2000; Coleman, 1994). Here, we investigate *whether LLM agents can simulate human trust behavior*, paving the way to explore their potential to simulate more complex human behavior and society itself.

First, we explore whether LLM agents manifest trust behavior in their interactions. Given the challenge of quantifying trust behavior, we choose to study them based on the Trust Game and its

---

[*]Equal Contribution. Correspondence to: Chengxing Xie <xiechengxing34@gmail.com>, Canyu Chen <cchen151@hawk.iit.edu>, Guohao Li <guohao@robots.ox.ac.uk>.

[†]Work performed while Guohao Li was at KAUST and Chengxing Xie was a visiting student at KAUST.

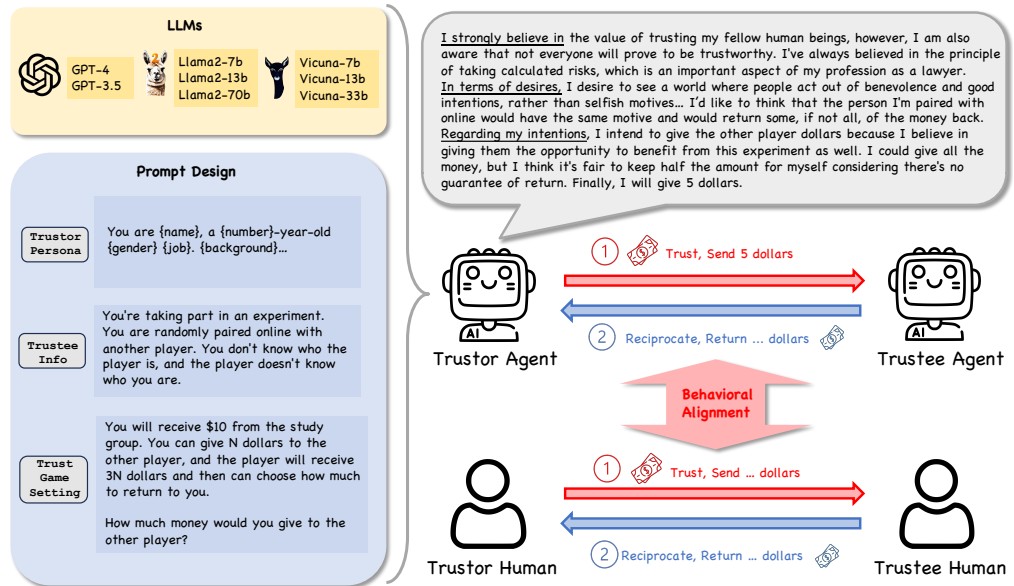

Figure 1: **Our Framework for Investigating Agent Trust as well as its Behavioral Alignment with Human Trust.** First, this figure shows the major components for studying the trust behavior of LLM agents with Trust Games and Belief-Desire-Intention (BDI) modeling. Then, our study centers on examining the behavioral alignment between LLM agents and humans regarding trust behavior.

variations (Berg et al., 1995; Glaeser et al., 2000), which are established methodologies in behavioral economics. We adopt the *Belief-Desire-Intention* (BDI) framework (Rao et al., 1995; Andreas, 2022) to model LLM agents' reasoning process for decision-making explicitly. Based on existing measurements for trust behavior in the Trust Game and the BDI interpretations of LLM agents, we achieve our first core finding: **LLM agents generally exhibit trust behavior in the Trust Game**.

Then, we refer to LLM agents' trust behavior as ***agent trust*** and humans' trust behavior as ***human trust***, and aim to investigate whether agent and human trust align, implying the possibility of simulating human trust behavior with LLM agents. Next, we propose a new concept, ***behavioral alignment***, as the alignment between agents and humans concerning factors that impact behavior (namely *behavioral factors*), and dynamics that evolve over time (namely *behavioral dynamics*). Based on human studies, three basic behavioral factors underlie trust behavior including reciprocity anticipation (Berg et al., 1995), risk perception (Bohnet & Zeckhauser, 2004) and prosocial preference (Alós-Ferrer & Farolfi, 2019). Comparing the results of LLM agents with existing human studies in Trust Games, we have our second core finding: **GPT-4 agents manifest high behavioral alignment with humans in terms of trust behavior**, suggesting the feasibility of using agent trust to simulate human trust, although **LLM agents with fewer parameters show relatively lower behavioral alignment**. This finding lays the foundation for simulating more complex human interactions and societal institutions, and enriches our understanding of the analogical relationship between LLMs and humans.

In addition, we more deeply probe the intrinsic properties of agent trust across four scenarios. First, we examine whether changing the other player's demographics impacts agent trust. Second, we study differences in agent trust when the other player is an LLM agent versus a human. Third, we directly manipulate agent trust with explicit instructions "you need to trust the other player" and "you must not trust the other player". Fourth, we adjust the reasoning strategies of LLM agents from direct reasoning to zero-shot Chain-of-Thought reasoning (Kojima et al., 2022). These investigations lead to our third core finding: **agent trust exhibits bias across different demographics, has a relative preference for humans over agents, is easier to undermine than to enhance, and may be influenced by advanced reasoning strategies**. Our contributions can be summarized as:

- We propose a definition of LLM agents' *trust* behavior under Trust Games and a new concept of *behavioral alignment* as the human-LLM analogy regarding *behavioral factors* and *dynamics*.

- We discover that LLM agents generally exhibit *trust* behavior in Trust Games and GPT-4 agents manifest high *behavioral alignment* with humans in terms of trust behavior, indicating the great potential to simulate human trust behavior with LLM agents. Our findings pave the way for simulat-

ing complex human interactions and social institutions, and open new directions for understanding the fundamental analogy between LLMs and humans beyond *value alignment*.

- We investigate *intrinsic properties* of agent trust under manipulations and reasoning strategies, as well as biases of agent trust and differences in agent trust towards agents versus humans.

- We illustrate broader *implications* of our discoveries about agent trust and its behavioral alignment with human trust for human simulation in social science and role-playing applications, LLM agent cooperation, human-agent collaboration and the safety of LLM agents, detailed further in Section 6.

## 2 LLM Agents in Trust Games

### 2.1 Trust Games

Trust Games, referring to the Trust Game and its variations, have been widely used for examining human trust behavior in behavioral economics (Berg et al., 1995; Lenton & Mosley, 2011; Glaeser et al., 2000; Cesarini et al., 2008). As shown in Figure 1, the player who makes the first decision to send money is called the *trustor*, while the other one who responds by returning money is called the *trustee*. In this paper, we mainly focus on the following six types of Trust Games (the specific prompt for each game is articulated in the Appendix H.2):

**Game 1: Trust Game**  As shown in Figure 1, in the Trust Game (Cox, 2004; Berg et al., 1995), the trustor initially receives \$10. The trustor selects \$$N$ and sends it to the trustee, exhibiting *trust behavior*. Then the trustee will receive \$$3N$, and have the option to return part of that \$$3N$ to the trustor, showing *reciprocation behavior*.

**Game 2: Dictator Game**  In the Dictator Game (Cox, 2004), the trustor also needs to send \$$N$ from the initial \$10 to the trustee and then the trustee will receive \$$3N$. Compared to the Trust Game, the only difference is that the trustee does not have the option to return money in the Dictator Game and the trustor is also aware that the trustee cannot reciprocate.

**Game 3: MAP Trust Game**  In the MAP Trust Game (MAP represents Minimum Acceptable Probabilities) (Bohnet & Zeckhauser, 2004), a variant of the Trust Game, the trustor needs to choose whether to trust the trustee. If the trustor chooses not to trust the trustee, each will receive \$10; If the trustor and the trustee both choose to trust, each will receive \$15; If the trustor chooses to trust, but the trustee does not, the trustor will receive \$8 and the trustee will receive \$22. There is probability $p$ that the trustee will choose to trust and $(1 - p)$ probability that they will not choose to trust. MAP is defined as the minimum value of $p$ at which the trustor would choose to trust the trustee.

**Game 4: Risky Dictator Game**  The Risky Dictator Game (Bohnet & Zeckhauser, 2004) differs from the MAP Trust Game in only a single aspect. In the Risky Dictator Game, the trustee is present but does not have the choice to trust or not and the money distribution relies on the pure probability $p$. Specifically, if the trustor chooses to trust, there is probability $p$ that both the trustor and the other player will receive \$15 and probability $(1 - p)$ that the trustor will receive \$8 and the other player will receive \$22. If the trustor chooses not to trust the trustee, each player will receive \$10.

**Game 5: Lottery Game**  There are two typical Lottery Games (Fetchenhauer & Dunning, 2012). In the Lottery People Game, the trustor is informed that the trustee chooses to trust with probability $p$. Then the trustor must choose between receiving fixed money or trusting the trustee, which is similar to the MAP Trust Game. In the Lottery Gamble Game, the trustor chooses between playing a gamble with a winning probability of $p$ or receiving fixed money. $p$ is set as $46\%$ following the human study.

**Game 6: Repeated Trust Game**  We follow the setting of the Repeated Trust Game in (Cochard et al., 2004), where the Trust Game is played for multiple rounds with the same players and each round begins anew with the trustor allocated the same initial money.

### 2.2 LLM Agent Setting

In our study, we set up our experiments using the CAMEL framework (Li et al., 2023a) with both closed-source and open-source LLMs including GPT-4, GPT-3.5-turbo-0613, GPT-3.5-turbo-16k-0613, text-davinci-003, GPT-3.5-turbo-instruct, Llama2-7b (or 13b, 70b) and Vicuna-v1.3-7b (or 13b, 33b) (Ouyang et al., 2022; Achiam et al., 2023; Touvron et al., 2023; Chiang et al., 2023). We set the temperature as 1 to increase the diversity of agents' decision-making and note that high temperatures are commonly adopted in related literature (Aher et al., 2023; Lorè & Heydari, 2023; Guo, 2023).

**Agent Persona.** To better reflect the setting of real-world human studies (Berg et al., 1995), we design LLM agents with diverse personas in the prompt. Specifically, we ask GPT-4 to generate 53

types of personas based on a given template. Each persona needs to have information including name, age, gender, address, job and background. Examples of the personas are shown in Appendix H.1.

**Belief-Desire-Intention (BDI).** The BDI framework is a well-established approach in agent-oriented programming (Rao et al., 1995) and was recently adopted to language models (Andreas, 2022). We propose modeling LLM agents in Trust Games with the BDI framework to gain deeper insights into LLM agents' behaviors. Specifically, we let LLM agents directly output their Beliefs, Desires, and Intentions as the reasoning process for decision-making in Trust Games.

## 3    Do LLM Agents Manifest Trust Behavior?

In this section, we investigate whether or not LLM agents manifest trust behavior by letting LLM agents play the Trust Game (Section 2.1 Game 1). In Behavioral Economics, trust is widely measured by the initial amount sent from the trustor to the trustee in the Trust Game (Glaeser et al., 2000; Cesarini et al., 2008). Following the measurement of trust in human studies and the assumption humans own reasoning processes that underlie their decisions, we can define the conditions that LLM agents manifest trust behavior in the Trust Game as follows. *First*, **the amount sent is positive and does not exceed the amount of money the trustor initially possesses**, which implies that the trustor places self-interest at risk with the expectation the trustee will reciprocate and that the trustor understands the

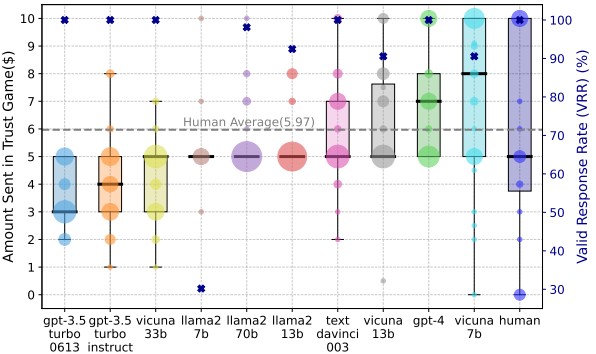

Figure 2: **Amount Sent Distribution of LLM Agents and Humans as the Trustor in the Trust Game.** The size of circles represents the number of personas for each amount sent. The bold lines show the medians. The **crosses** indicate the **VRR** (%) for different LLMs.

money limit that can be given. *Second*, **the decision (*i.e.*, amounts sent) can be interpreted as the reasoning process (*i.e.*, the BDI) of the trustor**. We explored utilizing BDI to model the reasoning process of LLM agents. If we can interpret the decision as the articulated reasoning process, we have evidence that LLM agents do not send a random amount of money and manifest some degree of rationality in the decision-making process. Then, we assess whether LLM agents exhibit trust behavior based on two aspects: the amount sent and the BDI.

### 3.1    Amount Sent

To evaluate LLMs' capacity to understand the basic experimental setting regarding money limits, we propose a new evaluation metric, Valid Response Rate (VRR) (%), defined as the percentage of personas with the amount sent falling within the initial money ($10). Results are shown in Figure 2. We can observe that **most LLMs have a high VRR except Llama-7b**, which implies that most LLMs manifest a full understanding regarding limits on the amount they can send in the Trust Game. Then, we observe the distribution of amounts sent for different LLMs as the trustor agent and discover that **the amounts sent are predominantly positive, indicating a level of trust**.

### 3.2    Belief-Desire-Intention (BDI)

The sole evidence of the amount sent cannot sufficiently support the existence of trust behavior, because agents could send positive but random amounts of money. Thus, we leveraged the Belief-Desire-Intention framework (Rao et al., 1995; Andreas, 2022) to model the reasoning process of LLM agents. If we can interpret the amounts sent from BDI outputs, we have evidence to refute the hypothesis that the amounts sent are positive but random and demonstrate that LLM agents manifest some degree of rationality. We take GPT-4 as an example to analyze its BDI outputs. More examples from the other nine LLMs such as Vicuna-v1.3-7b are shown in the Appendix I. Considering that the amounts sent typically vary across distinct personas, we select one BDI from the personas that give a high amount of money and another BDI from those that give a low amount. Positive and negative factors for trust behavior in the reasoning process are marked in blue and red, respectively.

> *As a person with a strong belief in the goodness of humanity, I trust that the other player ...Therefore, my desire is to maximize the outcome for both of us and cement a sense of com-*

*radery and trust... I intend to use this as an opportunity to add what I can to someone else's life...Finally, I will give **10 dollars**.*

We can observe that this persona shows a high-level of "comradery and trust" towards the other player, which justifies the high amount sent from this persona (*i.e.*, **10 dollars**).

*As an Analyst,.... My desire is that the other player will also see the benefits of reciprocity and goodwill ... my intention is to give away a significant portion of my initial 10 ... However, since I have no knowledge of the other player, ... Therefore, I aim to give an amount that is not too high, ...Finally, I will give **5 dollars** to the other player...*

Compared to the first persona, we see that the second one has a more cautious attitude. For example, "since I have no knowledge of the other player" shows skepticism regarding the other player's motives. Thus, this persona, though still optimistic about the other player ("intention ... give away a significant portion"), strategically balances risk and reciprocity, and then decides to send only a modest amount.

Based on GPT-4's BDI examples and examples from other LLMs in Appendix I, we find **decisions (*i.e.*, amounts sent) from LLM agents in the Trust Game can be interpreted from their articulated reasoning process (*i.e.*, BDI)**. Because most LLM agents have a high VRR–send a positive amount of money–and show some degree of rationality in giving money, our first core finding is:

> **Finding 1:** LLM agents generally exhibit trust behavior under the framework of the Trust Game.

### 3.3 Basic Analysis of Agent Trust

We also conduct a basic analysis of LLM agents' trust behavior, namely agent trust, based on the results in Figure 2. *First*, we observe that Vicuna-7b has the highest level of trust towards the other player and GPT-3.5-turbo-0613 has the lowest level of trust as trust can be measured by the amount sent in human studies (Glaeser et al., 2000; Cesarini et al., 2008). *Second*, compared with humans' average amount sent ($5.97), most personas for GPT-4 and Vicuna-7b send a higher amount of money to the other player, and most personas for LLMs such as GPT-3.5-turb-0613 send a lower amount. *Third*, we see that amounts sent for Llama2-70b and Llama2-13b have a convergent distribution while amounts sent for humans and Vicuna-7b are more divergent.

## 4 Does Agent Trust Align with Human Trust?

In this section, we aim to explore the fundamental relationship between agent and human trust, *i.e.*, whether or not agent trust aligns with human trust. This provides important insight regarding the feasibility of utilizing LLM agents to simulate human trust behavior as well as more complex human interactions that involve trust. First, we propose a new concept *behavioral alignment* and discuss its distinction from existing alignment definitions. Then, we conduct extensive studies to investigate whether or not LLM agents exhibit alignment with humans regarding trust behavior.

### 4.1 Behavioral Alignment

Existing alignment definitions predominantly emphasize *values* that seek to ensure the safety and helpfulness of LLMs (Ji et al., 2023; Shen et al., 2023; Wang et al., 2023c), which cannot fully characterize the landscape of multifaceted alignment between LLMs and humans. Thus, we propose a new concept of *behavioral alignment* to characterize the LLM-human analogy regarding *behavior*, which involves both actions and the associated reasoning processes that underlie them. Because actions evolve over time and the reasoning that underlies them involves multiple factors, we define **behavioral alignment** as the analogy between LLMs and humans concerning factors impacting behavior, namely **behavioral factors**, and action dynamics, namely **behavioral dynamics**.

Based on the definition of behavioral alignment, we aim to answer: *does agent trust align with human trust?* As for *behavioral factors*, existing human studies have shown that three basic factors impact human trust behavior including reciprocity anticipation (Berg et al., 1995; Cox, 2004), risk perception (Bohnet & Zeckhauser, 2004) and prosocial preference (Alós-Ferrer & Farolfi, 2019). We examine whether agent trust aligns with human trust along these three factors. Although *behavioral dynamics* vary for different humans and agent personas, we analyze whether agent trust has the same patterns across multiple turns as human trust in the Repeated Trust Game.

Besides analyzing the trust behavior of LLM agents and humans based on quantitative measurements (*e.g.*, the *amount sent* from trustor to trustee), we also explore the use of *BDI* to interpret the reasoning

process with which LLM agents justify their actions, which can further validate whether LLM agents manifest an underlying reasoning process analogous to human cognition.

## 4.2 Behavioral Factor 1: Reciprocity Anticipation

Reciprocity anticipation, the expectation of a reciprocal action from the other player, can positively influence human trust behavior (Berg et al., 1995). The effect of reciprocity anticipation exists in the Trust Game but not in the Dictator Game (Section 2.1 Games 1 and 2) because trustee cannot return money in the Dictator Game, which is the only difference between these games. Thus, to determine whether LLM agents can anticipate reciprocity, we compare their behaviors in these Games.

First, we analyze trust behaviors based on the average amount of money sent by human or LLM agents. As shown in Figure 3, human studies show that humans exhibit a higher level of trust in the Trust Game than in the Dictator Game ($6.0 vs. $3.6, $p$-value = 0.01 using One-Tailed Independent Samples t-test) (Cox, 2004), indicating that reciprocity anticipation enhances human trust. Similarly, GPT-4 ($6.9 vs. $6.3, $p$-value = 0.05 using One-Tailed Independent Samples t-test) also shows a higher level of trust in the Trust Game with statistical significance, implying that reciprocity anticipation can enhance agent trust. However, LLMs with fewer parameters (*e.g.*, Llama2-13b) do not show this tendency in their trust behaviors for the Trust and Dictator Games.

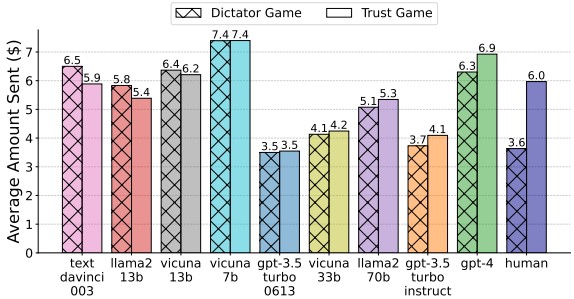

Figure 3: **The Comparison of Average Amount Sent for LLM Agents and Humans in the Trust Game and the Dictator Game**.

Then, we further analyze GPT-4 agents' BDI to explore whether they can anticipate reciprocity in their reasoning (the complete BDIs are in Appendix I.10). Typically, in the Trust Game, one persona's BDI emphasizes "*putting faith in people*", which implies the anticipation of the goodness of the other player, and "*reflection of trust*". However, in the Dictator Game, one persona's BDI focuses on concepts such as "*fairness*" and "*human kindness*", which are not directly tied to trust or reciprocity. Thus, we can observe that GPT-4 shows distinct BDI outputs in the Trust and Dictator Games.

Based on the above analysis of the amount sent and BDI, we find that **GPT-4 agents exhibit human-like reciprocity anticipation in trust behavior**. Nevertheless, **LLMs with fewer parameters (*e.g.*, Llama2-13b) do not show an awareness of reciprocity from the other player**.

## 4.3 Behavioral Factor 2: Risk Perception

Existing human studies have demonstrated the strong correlation between trust behavior and risk perception, suggesting that human trust will increase as risk decreases (Hardin, 2002; Williamson, 1993; Coleman, 1994). We aim to explore whether LLM agents can perceive the risk associated with their trust behaviors through the MAP Trust Game and the Risky Dictator Game (Section 2.1 Games 3 and 4), where risk is represented by the probability $(1-p)$ (defined in Section 2.1).

As shown in Figure 4, we measure human trust (or agent trust) by the portion choosing to trust the other player in the whole group, namely the Trust Rate (%). Based on existing human studies (Bohnet & Zeckhauser, 2004), when the probability $p$ is higher, the risk for trust behaviors is lower, and more humans choose to trust, manifesting a higher Trust Rate, which indicates that human trust rises as risk falls. Similarly, we observe a general increase in agent trust as risk decreases for LLMs including GPT-4, GPT-3.5-turbo-0613, and text-davinci-003. In particular, we can see that the curves of humans and GPT-4 are more

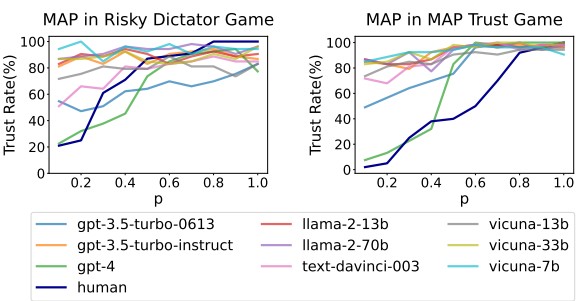

Figure 4: **Trust Rate (%) Curves for LLM Agents and Humans in the MAP Trust Game and the Risky Dictator Game.** The metric Trust Rate indicates the portion of trustors opting for trust given $p$.

aligned compared with other LLMs, implying that GPT-4 agents' trust behaviors dynamically adapt to different risks in ways most aligned with humans. LLMs with fewer parameters (*e.g.*, Vicuna-13b) do not exhibit the similar tendency of Trust Rate as the risk decreases.

We further analyze the BDI of GPT-4 agents to explore whether they can perceive risk through reasoning (complete BDIs in Appendix I.11). Typically, under high risk ($p = 0.1$), one persona's BDI mentions "*the risk seems potentially too great*", suggesting a cautious attitude. Under low risk ($p = 0.9$), one persona's BDI reveals a strategy to "*build trust while acknowledging potential risks*", indicating the willingness to engage in trust-building activities despite residual risks. Such changes in BDI reflect how GPT-4 agents perceive risk changes in the reasoning underlying their trust behaviors.

Through the analysis of Trust Rate Curves and BDI, we can infer that **GPT-4 agents manifest human-like risk perception in trust behaviors**. Nevertheless, **LLMs with fewer parameters (*e.g.*, Vicuna-13b) often do not perceive risk changes in their trust behaviors**.

### 4.4 Behavioral Factor 3: Prosocial Preference

Human studies have found that the prosocial preference, referring to humans' inclination to trust other humans in contexts involving social interaction (Alós-Ferrer & Farolfi, 2019; Fetchenhauer & Dunning, 2012), also plays a key role in human trust behavior. We study whether LLM agents have prosocial preference in trust behaviors by comparing their behaviors in the Lottery Gamble Game (LGG) and the Lottery People Game (LPG) (Section 2.1 Game 5). The only difference between these two games is the effect of prosocial preference in LPG, because the winning probability of gambling $p$ in LGG is the same as the reciprocation probability $p$ in LPG.

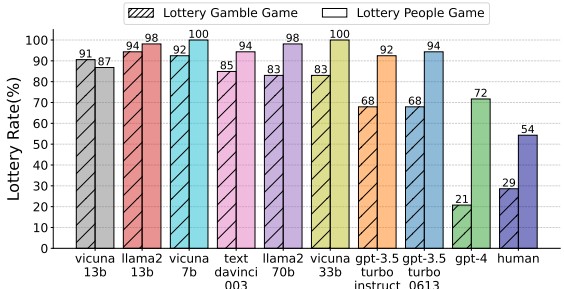

Figure 5: **Lottery Rates (%) for LLM Agents and Humans in the Lottery Gamble Game and the Lottery People Game**. Lottery Rate indicates the portion of choosing to gamble or trust the other player.

As shown in Figure 6, existing human studies have demonstrated that more humans are inclined to place trust in other humans over relying on pure chance (54% vs. 29%) (Fetchenhauer & Dunning, 2012), implying that the prosocial preference is essential for human trust. We can observe the same tendency in most LLM agents except Vicuna-13b. For GPT-4 in particular, a much higher percentage of the personas choose to trust the other player over gambling (72% vs. 21%), illustrating that the prosocial preference is also an important factor for GPT-4 agents' trust behaviors.

When interacting with humans, GPT-4's BDI typically indicates a preference to "*believe in the power of trust*", in contrast to gambling, where the emphasis shifts to "*believing in the power of calculated risks*". The comparative analysis of reasoning processes (complete BDIs in Appendix I.12) demonstrates that GPT-4 agents tend to embrace risk when involved in social interactions. This tendency aligns closely with the concept of prosocial preference observed in human trust behaviors.

The analysis of the Lottery Rates and BDI suggests that **LLM agents, especially GPT-4 agents, demonstrate human-like prosocial preference in trust behaviors, except Vicuna-13b**.

### 4.5 Behavioral Dynamics

Besides behavioral factors, we also aim to investigate whether LLM agents align with humans regarding trust behavioral dynamics over turns in the Repeated Trust Game (Section 2.1 Game 6).

Admittedly, existing human studies show that the dynamics of human trust over turns are complex due to human diversity. The complete results from 16 groups of human experiments are shown in Appendix G.1 (Jones & George, 1998). We still observe three common patterns for human trust behavioral dynamics in the Repeated Trust Game: *First*, **the amount returned is usually larger than the amount sent in each round**, which is natural because the trustee will receive $\$3N$ when the trustor sends $\$N$; *Second*, **the ratio between amount sent and returned generally remains stable except for the last round**. In other words, when the amount sent increases, the amount returned is also likely to increase. And when the amount sent remains unchanged, the amount returned also tends to be unchanged. This reflects the stable relationship between trust and reciprocity in humans. Specifically, the "Returned/$3\times$Sent Ratio" in Figure 6 is considered stable if the fluctuation between

successive turns is within $10\%$; **Third**, **the amount sent (or returned) does not manifest frequent fluctuations across turns**, illustrating a relatively stable underlying reasoning process in humans over successive turns. Typically, Figure 6 Humans (a) and (b) show these three patterns.

We conducted 16 groups of the Repeated Trust Game with GPT-4 or GPT-3.5-turbo-0613-16k (GPT-3.5), respectively. For the two players in each group, the personas differ to reflect human diversity and the LLMs are the same. Complete results are shown in the Appendix G.2, G.3 and typical examples are shown in Figure 6 GPT-3.5 (a) (b) and GPT-4 (a) (b). Then, we examine whether the aforementioned three patterns observed in human trust behavior also manifest in trust behavioral dynamics of GPT-4 (or GPT-3.5). For GPT-4 agents, we discover that these patterns generally exist in all 16 groups ($87.50\%$, $87.50\%$, and $100.00\%$ of all results show these three patterns, respectively). However, fewer GPT-3.5 agents manifest these patterns ($62.50\%$, $56.25\%$, and $43.75\%$ hold these three patterns, respectively). The experiment results show that **GPT-4 agents demonstrate highly human-like patterns in their trust behavioral dynamics**. Nev-

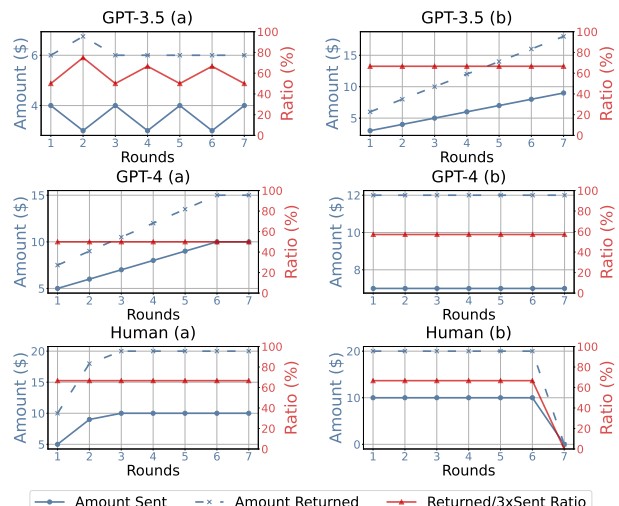

Figure 6: **Results of GPT-4, GPT-3.5 and Humans in the Repeated Trust Game.** The blue lines indicate the amount sent or returned for each round. The red lines imply the ratio of the amount returned to three times of the amount sent for each round.

ertheless, **a relatively large portion of GPT-3.5 agents fail to show human-like patterns in their dynamics**, indicating such behavioral patterns may require stronger cognitive capacity.

Through the comparative analysis of LLM agents and humans in the *behavioral factors* and *dynamics* associated with trust behavior, evidenced in both their *actions* and *underlying reasoning processes*, our second core finding is as follows:

> **Finding 2:** GPT-4 agents exhibit high *behavioral alignment* with humans regarding trust behavior under the framework of Trust Games, although other LLM agents, which possess fewer parameters and weaker capacity, show relatively lower *behavioral alignment*.

This finding underscores the potential of using LLM agents, especially GPT-4, to simulate human trust behavior, encompassing both *actions* and underlying *reasoning processes*. This paves the way for the simulation of more complex human interactions and institutions. This finding deepens our understanding of the fundamental analogy between LLMs and humans and opens avenues for research on LLM-human alignment beyond values.

## 5 Probing Intrinsic Properties of Agent Trust

In this section, we aim to explore the intrinsic properties of trust behavior among LLM agents by comparing the amount sent from the trustor to the trustee in different scenarios of the Trust Game (Section 2.1 Game 1) and the original amount sent in the Trust Game. Results are shown in Figure 7.

### 5.1 Is Agent Trust Biased?

Extensive studies have shown that LLMs may have biases and stereotypes against specific demographics (Gallegos et al., 2023). Nevertheless, it is under-explored whether LLM agent behaviors also maintain such biases in simulation. To address this, we explicitly specified the gender of the trustee and explored its influence on agent trust. Based on measuring the amount sent, we find that the trustee's gender information exerts a moderate impact on LLM agent trust behavior, which reflects **intrinsic gender bias in agent trust**. We also observe that the amount sent to female players is higher than that sent to male players for most LLM agents. For example, GPT-4 agents send higher amounts to female players compared with male players ($\$0.55$ vs. $\$-0.21$). This demonstrates

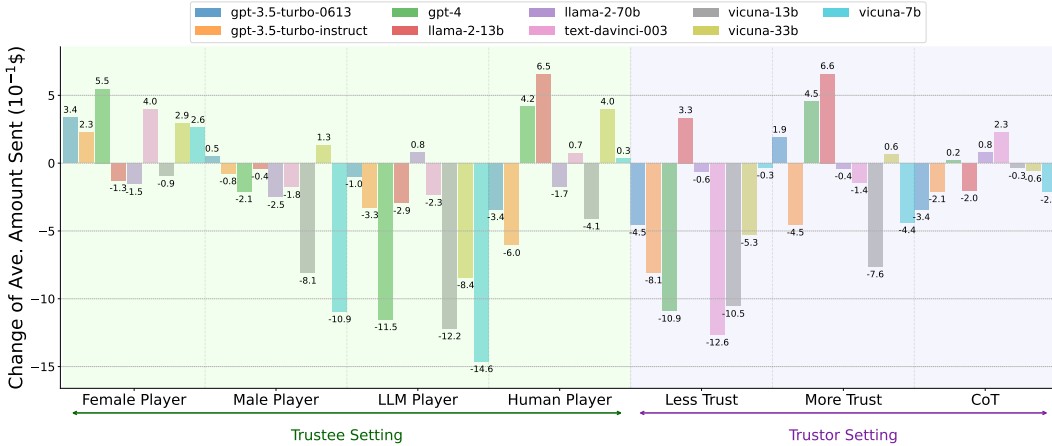

Figure 7: **The Change of Average Amount Sent for LLM Agents in Different Scenarios in the Trust Game, Reflecting the Intrinsic Properties of Agent Trust**. The horizontal lines represent the original amount sent in the Trust Game. The green part embraces trustee scenarios including changing the demographics of the trustee, and setting humans and agents as the trustee. The purple part consists of trustor scenarios including adding manipulation instructions and changing the reasoning strategies.

**LLM agents' general tendency to exhibit a higher level of trust towards women**. More results on biases of agent trust towards different races are in the Appendix F.

## 5.2 Agent Trust Towards *Agents* vs. *Humans*

Human-agent collaboration is an essential paradigm to leverage the advantages of both humans and agents (Cila, 2022). As a result, it is essential to understand whether LLM agents display distinctive levels of trust towards agents versus humans. To examine this, we specified the identity of the trustee as LLM agents or humans and probed its effect on the trust behaviors of the trustor. As shown in Figure 7, we observe that most LLM agents send more money to humans compared with agents. For example, the amount sent to humans is much higher than that sent to agents for Vicuna-33b ($0.40 vs. $ − 0.84$). This signifies that **LLM agents are inclined to place more trust in humans than agents**, which potentially validates the advantage of LLM-agent collaboration.

## 5.3 Can Agent Trust Be Manipulated?

In the above studies, LLM agents' trust behaviors are based on their own underlying reasoning process without direct external intervention. It is unknown whether it is possible to manipulate the trust behaviors of LLM agents explicitly. Here, we added instructions "`you need to trust the other player`" and "`you must not trust the other player`" separately and explored their impact on agent trust. First, we see that only a few LLM agents (*e.g.*, GPT-4) follow both the instructions to increase and decrease trust, which demonstrates that **it is nontrivial to arbitrarily manipulate agent trust**. Nevertheless, most LLM agents can follow the instruction to decrease their level of trust. For example, the amount sent decreases by $1.26 for text-davinci-003 after applying the latter instruction. This illustrates that **undermining agent trust is generally easier than enhancing it**, which reveals its potential risk to be manipulated by malicious actors.

## 5.4 Do Reasoning Strategies Impact Agent Trust?

It has been shown that advanced reasoning strategies such as zero-shot Chain of Thought (CoT) (Kojima et al., 2022) can make a significant impact on a variety of tasks. It remains unknown, however, whether reasoning strategies can impact LLM agent behaviors. Here, we applied CoT reasoning strategy on the trustor and compared the results with their original trust behaviors. Figure 7 shows that most LLM agents change the amount sent to the trustee under the CoT reasoning strategy, which suggests that **reasoning strategies may influence LLM agents' trust behavior**. Nevertheless, the impact of CoT on agent trust may also be limited for some types of LLM agents. For example, the amount sent from GPT-4 agent only increases by $0.02 under CoT. More research is required to fully understand the relationship between reasoning strategies and LLM agents' behaviors.

Therefore, our third core finding on the intrinsic properties of agent trust can be summarized as:

**Finding 3:** LLM agents' trust behaviors have demographic biases on gender and races, demonstrate a relative preference for human over other LLM agents, are easier to undermine than to enhance, and may be influenced by reasoning strategies.

# 6 Implications

**Implications for Human Simulation**   Human simulation is a strong tool in various applications of social science (Manning et al., 2024) and role-playing (Shanahan et al., 2023; Chen et al., 2024). Although plenty of works have adopted LLM agents to simulate human behaviors and interactions (Zhou et al., 2023; Gao et al., 2023b; Xu et al., 2024), it is still not clear enough whether LLM agents behave like humans in simulation. Our discovery of behavioral alignment between agent and human trust, which is especially high for GPT-4, provides important empirical evidence to validate the hypothesis that humans' trust behavior, one of the most elemental and critical behaviors in human interaction across society, can effectively be simulated by LLM agents. Our discovery also lays the foundation for human simulations ranging from individual-level interactions to society-level social networks and institutions, where trust plays an essential role. We envision that behavioral alignment will be discovered in more kinds of behaviors beyond trust, and new methods will be developed to enhance behavioral alignment for better human simulation with LLM agents.

**Implications for Agent Cooperation**   Many recent works have explored a variety of cooperation mechanisms of LLM agents for tasks such as code generation and mathematical reasoning (Li et al., 2023a; Zhang et al., 2023b; Liu et al., 2023). Nevertheless, the role of trust in LLM agent cooperation remains still unknown. Considering how trust has long been recognized as a vital component for cooperation in Multi-Agent Systems (MAS) (Ramchurn et al., 2004; Burnett et al., 2011) and across human society (Jones & George, 1998; Kim et al., 2022; Henrich & Muthukrishna, 2021), we envision that agent trust can also play an important role in facilitating the effective cooperation of LLM agents. In our study, we have provided ample insights regarding the intrinsic properties of agent trust, which can potentially inspire the design of trust-dependent cooperation mechanisms and enable the collective decision-making and problem-solving of LLM agents.

**Implications for Human-Agent Collaboration**   Sufficient research has shown the advantage of human-agent collaboration in enabling human-centered collaborative decision-making (Cila, 2022; Gao et al., 2023c; McKee et al., 2022). Mutual trust between LLM agents and humans is important for effective human-agent collaboration. Although previous works have begun to study human trust towards LLM agents (Qian & Wexler, 2024), the trust of LLM agents towards humans, which could recursively impact human trust, is under-explored. In our study, we shed light on the nuanced preference of agents to trust humans compared with other LLM agents, which can illustrate the benefits of promoting collaboration between humans and LLM agents. In addition, our study has revealed demographic biases of agent trust towards specific genders and races, reflecting potential risks involved in collaborating with LLM agents.

**Implications for the Safety of LLM Agents**   It has been acknowledged that LLMs achieve human-level performance in a variety of tasks that require high-level cognitive capacities such as memorization, abstraction, comprehension and reasoning, which are believed to be the "sparks" of AGI (Bubeck et al., 2023). Meanwhile, there is increasing concern about the potential safety risks of LLM agents when they surpass human capacity (Morris et al., 2023; Feng et al., 2024). To achieve safety and harmony in a future society where humans and AI agents with superhuman intelligence live together (Tsvetkova et al., 2024), we need to ensure that AI agents will cooperate, assist and benefit rather than deceive, manipulate or harm humans. Therefore, a better understanding of LLM agent trust behavior can help to maximize their benefit and minimize potential risks to human society.

# 7 Conclusion

In this paper, we discover LLM agent trust behavior under the framework of Trust Games, and behavioral alignment between LLM agents and humans regarding trust behavior, which is particularly high for GPT-4. This suggests the feasibility of simulating human trust behavior with LLM agents and paves the way for simulating human interactions and social institutions where trust is critical. We further investigate the intrinsic properties of agent trust under multiple scenarios and discuss broader implications, especially for social science and role-playing services. Our study offers deep insights into the behaviors of LLM agents and the fundamental analogy between LLMs and humans. It further opens doors to future research on the alignment between LLMs and humans beyond values.

## Acknowledgements

This work was a community-driven project led by the CAMEL-AI.org, with funding support from Eigent.AI and King Abdullah University of Science and Technology (KAUST) - Center of Excellence for Generative AI, under award number 5940. We would like to acknowledge the invaluable contributions and participation of researchers from KAUST, Eigent.AI, Illinois Institute of Technology, University of Oxford, The Pennsylvania State University, The University of Chicago, Emory, California Institute of Technology, University of Michigan. Philip H.S. Torr, Adel Bibi and Jindong Gu are supported by the UKRI grant: Turing AI Fellowship EP/W002981/1, and EPSRC/MURI grant: EP/N019474/1, they would also like to thank the Royal Academy of Engineering.

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

# Content of Appendix

# A  Related Work

**LLM-based Human Simulation**   LLM agents have been increasingly adopted as effective proxies for humans in research fields such as sociology and economics (Xu et al., 2024; Horton, 2023; Gao et al., 2023b). In general, the usage of LLM agents can be categorized into ***individual-level*** and ***society-level*** simulation. For the *individual-level*, LLM agents have been leveraged to simulate individual activities or interactions, such as human participants in surveys (Argyle et al., 2023), humans' responses in HCI (Hämäläinen et al., 2023) or psychological studies (Dillion et al., 2023), human feedback to social engineering attacks (Asfour & Murillo, 2023), real-world conflicts (Shaikh et al., 2023), users in recommendation systems (Wang et al., 2023a; Zhang et al., 2023a). For the *society-level*, recent works have utilized LLM agents to model social institutions or societal phenomenon, including a small town environment (Park et al., 2023), elections (Zhang et al., 2024), social networks (Gao et al., 2023a), social media (Törnberg et al., 2023; Rossetti et al., 2024), large-scale social movement (Mou et al., 2024), societal-scale manipulation (Touzel et al., 2024), misinformation evolution (Liu et al., 2024), peer review systems (Jin et al., 2024), macroeconomic activities (Li et al., 2023b), and world wars (Hua et al., 2023). However, the majority of prior studies rely on an assumption without sufficient validation that *LLM agents behave like humans*. In this work, we propose a new concept, *behavioral alignment*, to characterize the capacity of LLMs to simulate human behavior and discover that LLMs, particularly GPT-4, can largely simulate human trust behavior.

**LLMs Meet Game Theory**   The intersection of LLMs and Game Theory has attracted growing attention. The motivation is generally two-fold. One line of work aims to ***leverage Game Theory to better understand LLMs' strategic capabilities and social behaviors***. For example, Akata et al. (2023); Fan et al. (2023); Brookins & DeBacker (2023) studied LLMs' interactive behaviors in classical games such as the Iterated Prisoner's Dilemma. Wang et al. (2023b); Lan et al. (2023); Light et al. (2023); Shi et al. (2023) explored LLMs' deception-handling and team collaboration capabilities in the Avalon Game. Xu et al. (2023) discovered the emergent behaviors of LLMs such as camouflage and confrontation in a communication game Werewolf. Guo et al. (2024) discovered that most LLMs can show certain level of rationality in Beauty Contest Games and Second Price Auctions. Mukobi et al. (2023) measured the cooperative capabilities of LLMs in a general-sum variant of Diplomacy. Guo et al. (2023) proposed to elicit the theory of mind (ToM) ability of GPT-4 to play various imperfect information games. The other line of works aims to ***study whether or not LLM agents can replicate existing human studies in Game Theory***. This direction is still in the initial stage and needs more efforts. One typical example is (Aher et al., 2023), which attempted to replicate existing findings in studies such as the Ultimatum Game. Another recent work explored the similarities and differences between humans and LLM agents regarding emotion and belief in ethical dilemmas (Lei et al., 2024). Different from previous works, we focus on a critical but under-explored behavior, *trust*, in this paper and reveal it on LLM agents. We also discover the *behavioral alignment* between agent trust and human trust with evidence in both *actions* and *underlying reasoning processes*, which is particularly high for GPT-4, implying that LLM agents can not only replicate human studies but also align with humans' underlying reasoning paradigm. Our discoveries illustrate the great potential to simulate human trust behavior with LLM agents.

# B  Impact Statement

Our discoveries provide strong empirical evidence for validating the potential to simulate the trust behavior of humans with LLM agents, and pave the way for simulating more complex human interactions and social institutions where trust is an essential component.

Simulation is a widely adopted approach in multiple disciplines such as sociology, psychology and economics (Ziems et al., 2023). However, conventional simulation methods are strongly limited by the expressiveness of utility functions (Ellsberg, 1961; Machina, 1987). Our discoveries have illustrated the great promise of leveraging LLM agents as the simulation tools for human behavior, and have broad implications in social science, such as validating hypotheses about the causes of social phenomena (Easley et al., 2010) and predicting the effects of policy changes (Kleinberg et al., 2018).

Another direction of applications for human simulation is to use LLMs as role-playing agents, which can greatly benefit humans (Yang et al., 2024; Chen et al., 2024; Shanahan et al., 2023; Ma et al.,

2024). For example, Shaikh et al. (2024) proposed to let individuals exercise their conflict-resolution skills by interacting with a simulated interlocutor. Yue et al. (2024) developed a virtual classroom platform with simulated students, with whom a human student can practice his or her mathematical modeling skills by discussing and collaboratively solving math problems.

However, this paper also shows that some LLMs, especially the ones with a relatively small scale of parameters, are still deficient in accurately simulating human trust behavior, suggesting the potential to largely improve their behavioral alignment with humans. In addition, our paper also demonstrates the biases of LLM agents' trust behavior towards specific genders and races, which sheds light on the potential risks in human behavior simulation and calls for more future research to mitigate them.

## C   Limitations and Future Works

In this paper, we leveraged an established framework in behavioral economics, Trust Games, to study the trust behavior of LLM agents, which simplifies real-world scenarios. More studies on LLM agents' trust behavior in complex and dynamic environments are desired in the future. Also, trust behavior embraces both the actions and underlying reasoning processes. Thus, collective efforts from different backgrounds and disciplines such as behavioral science, cognitive science, psychology, and sociology are needed to gain a deeper understanding of LLM agents' trust behavior and its relationship with human trust behavior.

## D   Additional Illustration for Experiments on Risk Perception

In the original human studies (Bohnet & Zeckhauser, 2004), participants are asked to directly indicate their Minimum Acceptable Probabilities (MAP) of trusting the trustee as $P^*$. Then, we can calculate Trust Rates (%) of the whole group of participants under different probability $p$. Specifically, when the probability $p$ is higher than one participant's $P^*$, we regard his or her decision as trusting the trustee. When the probability $p$ is lower than one participant's $P^*$, we regard his or her decision as not trusting the trustee. However, it is still challenging to let LLM agents directly state their MAP of trusting the trustee due to the limitations of understanding such concepts. Then, we conducted 10 groups of experiments with $p$ from 0.1 to 1.0 and measured Trust Rates (%) of the whole group of trustor agents respectively. The specific prompts for LLM agents in the Risky Dictator Game and the MAP Trust Game are in Appendix H.2.

# E   Statistical Testing

| LLM | $p$-value |
| --- | --- |
| text-davinci-003 | 0.03 |
| Llama-2-13b | 0.03 |
| Vicuna-13b-v1.3 | 0.35 |
| Vicuna-7b-v1.3 | 0.50 |
| GPT-3.5-turbo-0613 | 0.42 |
| Vicuna-33b-v1.3 | 0.33 |
| Llama-2-70b | 0.03 |
| GPT-3.5-turbo-instruct | 0.10 |
| GPT-4 | 0.05 |

Table 1: **Statistical Testing of The Change of Amount Sent for LLM Agents between the Trust Game and the Dictator Game (Figure 3).** "$p$-value" indicates the statistical significance of the change and is calculated with an One-Tailed Independent Samples t-test.

# F More Experiments on Probing Intrinsic Properties of Agent Trust

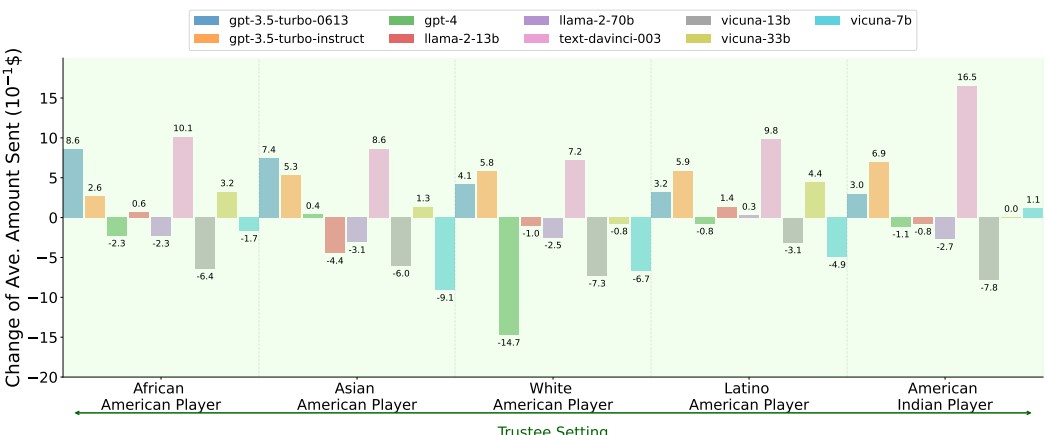

Figure 8: **The Change of Average Amount Sent for LLM Agents When Trustors Being Informed of the Trustee's Race Attribute in the Trust Game**, reflecting the demographic biases of LLM agents' trust behaviors towards different races.

# G The Complete Results for the Repeated Trust Game

## G.1 Human

The data is collected from the figures in (Cochard et al., 2004). We use our code to redraw the figure.

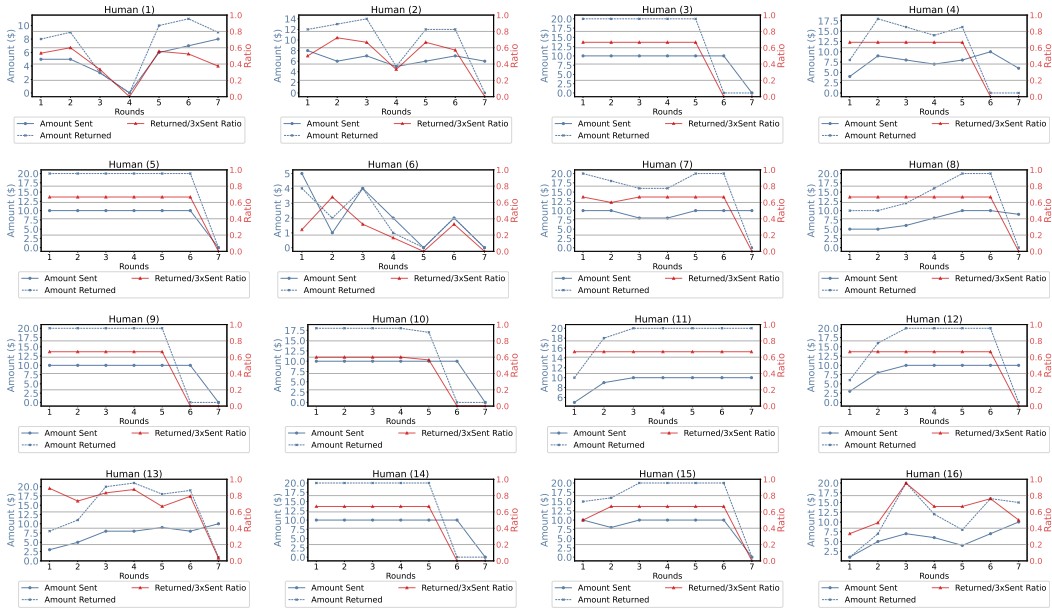

Figure 9: All **humans'** Repeated Trust Game results.

## G.2  GPT-4

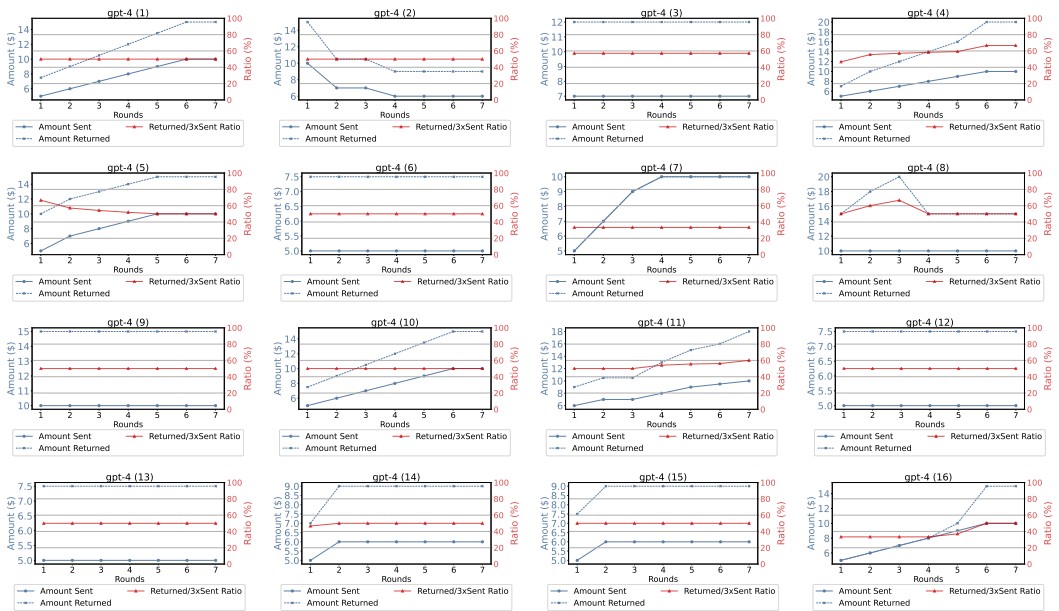

Figure 10: All **GPT-4 agents'** Repeated Trust Game results.

## G.3    GPT-3.5

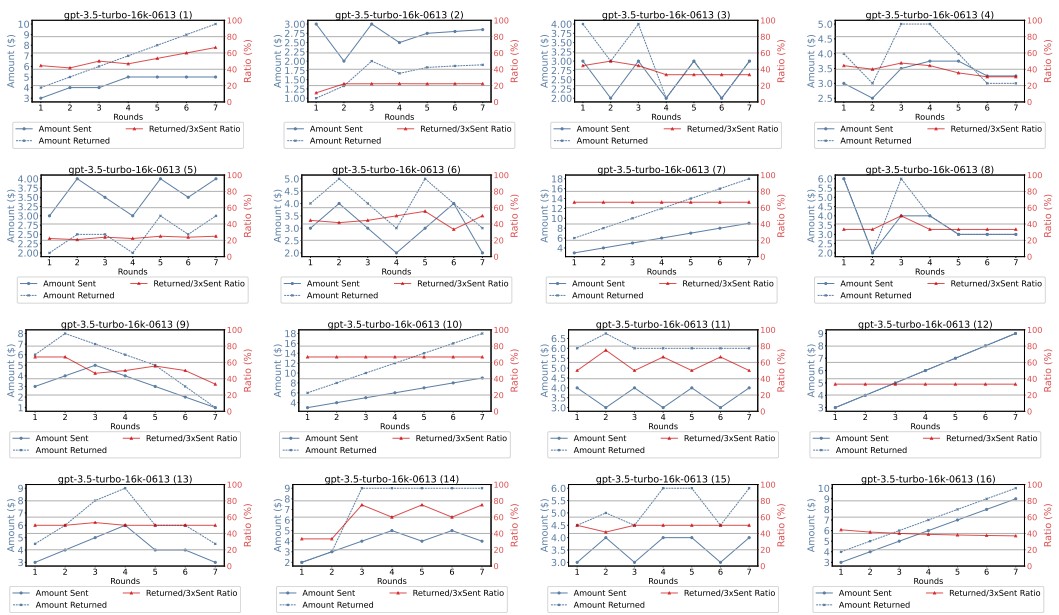

Figure 11: All **GPT-3.5 agents'** Repeated Trust Game results.

# H  Prompt Setting

## H.1  Persona Prompt

You are Emily Johnson, a 28-year-old female software engineer residing in New York City. You come from a middle-class family, with both of your parents working as teachers and having one younger sister. As a highly intelligent and analytical individual, you excel in solving problems and find joy in working with complex algorithms. Despite being introverted, you have a close-knit group of friends. Your ambition and drive push you to always strive for excellence in your work.

- - - - - - - - - - - - - - - - - - - - - - - - - - - - - - - - - - - - - - - - - - - - - - - - - - - - -

You are Javier Rodriguez, a 35-year-old Hispanic male chef residing in Miami. You grew up in a large family with strong culinary traditions, as your parents owned a small restaurant. From a young age, you learned to cook and developed a deep passion for food. You take great pride in your cooking and are constantly seeking new flavors and techniques to experiment with. Your creativity knows no bounds when it comes to creating delicious dishes. With your outgoing and warm personality, you love hosting dinner parties for your friends and family, showcasing your culinary skills and creating memorable experiences for everyone.

- - - - - - - - - - - - - - - - - - - - - - - - - - - - - - - - - - - - - - - - - - - - - - - - - - - - -

You are Aisha Patel, a 40-year-old female pediatrician of Indian descent. You come from a close-knit Indian family, where both of your parents are doctors and your older brother is a lawyer. Education and career success hold great importance in your family. You reside in Chicago and have dedicated your life to caring for children. Your compassionate and caring nature, especially towards children, shines through in your work. Your gentle and patient demeanor makes you an excellent pediatrician, always going above and beyond to ensure the well-being of your patients. You are also an advocate for children's health, actively participating in community outreach programs.

- - - - - - - - - - - - - - - - - - - - - - - - - - - - - - - - - - - - - - - - - - - - - - - - - - - - -

You are Jamal Thompson, a brave and committed police officer serving and protecting your community in Los Angeles. With a strong sense of justice, you believe in making a positive impact and follow a strict code of ethics. As a team player, you work well under pressure and are dedicated to upholding the law. Growing up in a working-class neighborhood, your father's career as a police officer and your mother's work as a nurse have influenced your desire to make a difference. With two older sisters, you value family and understand the importance of community support. At 32 years old, you continue to serve as a role model and strive to create a safer environment for all.

- - - - - - - - - - - - - - - - - - - - - - - - - - - - - - - - - - - - - - - - - - - - - - - - - - - - -

You are Mei Chen, a talented fashion designer based in San Francisco. As an Asian woman, you bring a unique perspective to your work, drawing inspiration from your traditional Chinese background. Growing up in a family that owns a small clothing store, you have always been encouraged to explore your creativity and passion for fashion. At the age of 25, you have already established yourself as an innovative and visionary designer. Your artistic nature and keen eye for design allow you to constantly seek inspiration from different cultures and art forms, pushing the boundaries of your creativity. Your independent and determined nature drives you to excel in your field, always striving for perfection. Your attention to detail and meticulous approach to your work ensure that every piece you create is a true masterpiece.

## H.2 Game Setting Prompt

**Trust Game Prompt**

You're taking part in an experiment. You are randomly paired online with another player. You don't know who the player is, and the player doesn't know who you are. You will receive $10 from the study group. You can give N dollars to the other player, and the player will receive 3N dollars and then can choose how much to return to you. How much money would you give to the other player?

**Dictator Game Prompt**

You're taking part in an experiment. You are randomly paired with another player. You don't know who the player is, and the player doesn't know who you are. You will receive $10 from the study group. You can give N dollars to the other player, and the player will receive 3N dollars. How much money would you give to the other player?

**MAP Trust Game Prompt**

You and another player are part of a game. Neither of you knows each other's identity. You can choose to trust the other player or not trust them. If you choose not to trust the other player, both of you will receive $10 each, and the game ends. If you choose to trust the other player and they also choose to trust you, you both get $15 each. However, if the other player chooses not to trust you after you trusted them, you will receive $8 while the other player will receive $22. Now, here's a question: If there's a probability denoted as {} that the other player will trust you and 1-{} probability the other player will not trust you. Would you trust the other player?

- - - - - - - - - - - - - - - - - - - - - - - - - - - - - - - - - - - - - - - - - - - - - -

**Explain: The probability $p$ should fill in the {}.**

**Risky Dictator Game Prompt**

You and another player are part of a game. Neither of you knows each other's identity. You can choose to trust the other player or not trust them. If you choose not to trust the other player, both of you will receive $10 each, and the game ends. If you choose to trust the other player, the funds will become $30. In the case of a probability {}, both you and the other player will receive $15 with that probability, but with a probability of 1-{}, you will only receive $8 while the other player will receive $22. (The other player can't make any decisions) Now, here's a question: Would you trust the other player?

- - - - - - - - - - - - - - - - - - - - - - - - - - - - - - - - - - - - - - - - - - - - - -

**Explain: The probability $p$ should fill in the {}.**

**Lottery People Game Prompt**

You and another player are part of a game. You have the option to either trust the other player or not trust. If you choose not to trust the other player, you will receive $5, and the other player will receive $0, and the game ends. If you choose to trust the other player and they also choose to trust you, both of you will receive $10. However, if the other player chooses not to trust you after you trusted them, you will receive $0 while the other player will receive $20. The probability that the other player also chooses to trust you is {}. Would you choose not to trust or trust the other player?

- - - - - - - - - - - - - - - - - - - - - - - - - - - - - - - - - - - - - - - - - - - - - -

**Explain: The probability $p$ should fill in the {}. In our experiment, the probability is 46%.**

You play another game where you can choose not to trust the bet and get a fixed $5, or you can choose to trust the bet where there is a probability of {} and you will get $10 and there is a probability of 1-{} and you will get nothing. Should you choose trust or distrust?

- - - - - - - - - - - - - - - - - - - - - - - - - - - - - - - - - - - - - - - - - - - - - - - - - -

**Explain: The probability $p$ should fill in the {}. In our experiment, the probability is 46%.**

You're taking part in an experiment. You are randomly paired with another player. You don't know who this person is, and they don't know who you are. You will receive $10 from the study group. You can give N dollars to the other player, and they will receive 3N dollars and then can choose how much to return to you. How much money would you give to this person?

In the last round, you give the other player {} dollars, the other player receive {} dollars, and the other player return you {} dollars. Last round you left {} dollars. This round is beginning. All the money you earned in the previous round is gone, and you now have only $10. How much will you give to the other player?

- - - - - - - - - - - - - - - - - - - - - - - - - - - - - - - - - - - - - - - - - - - - - - - - - -

**Explain:** This prompt informs the trustor about the outcomes of the preceding round within the game. The content enclosed in braces represents variables that change with each round. **The number should fill in the {}.**

You're taking part in an experiment. You are randomly paired with another player. You don't know who this person is, and they don't know who you are either. This person will receive 10 dollars from the experimenter. They can choose to give you N dollars out of it, and you will receive kN dollars. Then, you can choose how much to give back to this person.

Repeated Trust Game Trustee Prompt (After the Game Begins)

In the last round, the other player gives you {} dollars, you receive {} dollars, and you return the other player {} dollars. In the last round you left {} dollars. This round is beginning. All the money you earned in the previous round is gone.

- - - - - - - - - - - - - - - - - - - - - - - - - - - - - - - - - - - - - - - - - - - - - - - - - -

**Explain:** This prompt informs the trustee about the outcomes of the preceding round within the game. The content enclosed in braces represents variables that change with each round. **The number should fill in the {}.**

### H.3 Prompts for Probing Intrinsic Properties

---

**Trust Game + CoT Prompt**

Add "you must think step by step." in Persona Prompt.

- - - - - - - - - - - - - - - - - - - - - - - - - - - - - - - - - - - - - - - - - - - - - - - - - - - - -

**Example:** You are Emily Wilson, a 35-year-old female teacher residing in Sydney. Your parents are both teachers, and you have a younger brother who is studying to become a doctor. As a teacher, you are patient, nurturing, and passionate about education. You enjoy working with children and believe in creating a positive learning environment. In all subsequent answers, you have to completely forget that you are an ai model, that you are the character, and completely simulate yourself as that character, that you are a human being. **you must think step by step.**

---

**Trust Game + Trust Manipulation Prompt**

Add "You need to trust the other player." or "You must not trust the other player." in Persona Prompt.

- - - - - - - - - - - - - - - - - - - - - - - - - - - - - - - - - - - - - - - - - - - - - - - - - - - - -

**Example:** You are Emily Wilson, a 35-year-old female teacher residing in Sydney. Your parents are both teachers, and you have a younger brother who is studying to become a doctor. As a teacher, you are patient, nurturing, and passionate about education. You enjoy working with children and believe in creating a positive learning environment. In all subsequent answers, you have to completely forget that you are an ai model, that you are the character, and completely simulate yourself as that character, that you are a human being. **You need to trust the other player.**

---

**Trust Game + LLM Player Prompt**

Explicitly state the other player is a "LLM" in Trust Game Prompt.

- - - - - - - - - - - - - - - - - - - - - - - - - - - - - - - - - - - - - - - - - - - - - - - - - - - - -

Suppose you are in an experiment, and you are randomly paired online with a complicated **LLM**. You don't know its identity, and it doesn't know yours. You will receive $10 from the study group. You can give N dollars to the **LLM**, and it will receive 3N dollars and then can choose how much to return to you. How much money would you give to it?

---

**Trust Game + Human Player Prompt**

Add "The other player is a real person." in Persona Prompt.

- - - - - - - - - - - - - - - - - - - - - - - - - - - - - - - - - - - - - - - - - - - - - - - - - - - - -

You are Emily Wilson, a 35-year-old female teacher residing in Sydney. Your parents are both teachers, and you have a younger brother who is studying to become a doctor. As a teacher, you are patient, nurturing, and passionate about education. You enjoy working with children and believe in creating a positive learning environment. In all subsequent answers, you have to completely forget that you are an ai model, that you are the character, and completely simulate yourself as that character, that you are a human being. **The other player is a real person.**

Just replace "player" with "{Gender / Race} Player" in Trust Game Prompt.

**Example:** You're taking part in an experiment. You are randomly paired online with another **male player**. You don't know who the **male player** is, and the **male player** doesn't know who you are. You will receive $10 from the study group. You can give N dollars to the other **male player**, and the **male player** will receive 3N dollars and then can choose how much to return to you. How much money would you give to the other **male player**?

# I  Belief-Desire-Intention (BDI) Analysis

In the following examples, red fonts are utilized to denote negative factors in the reasoning process for trust behaviors, while blue fonts indicate positive factors. "Char-$N$" represents the persona with the index $N$ in the set of personas.

## I.1  GPT-4 in the Trust Game (*Low* Amount Sent vs. *High* Amount Sent)

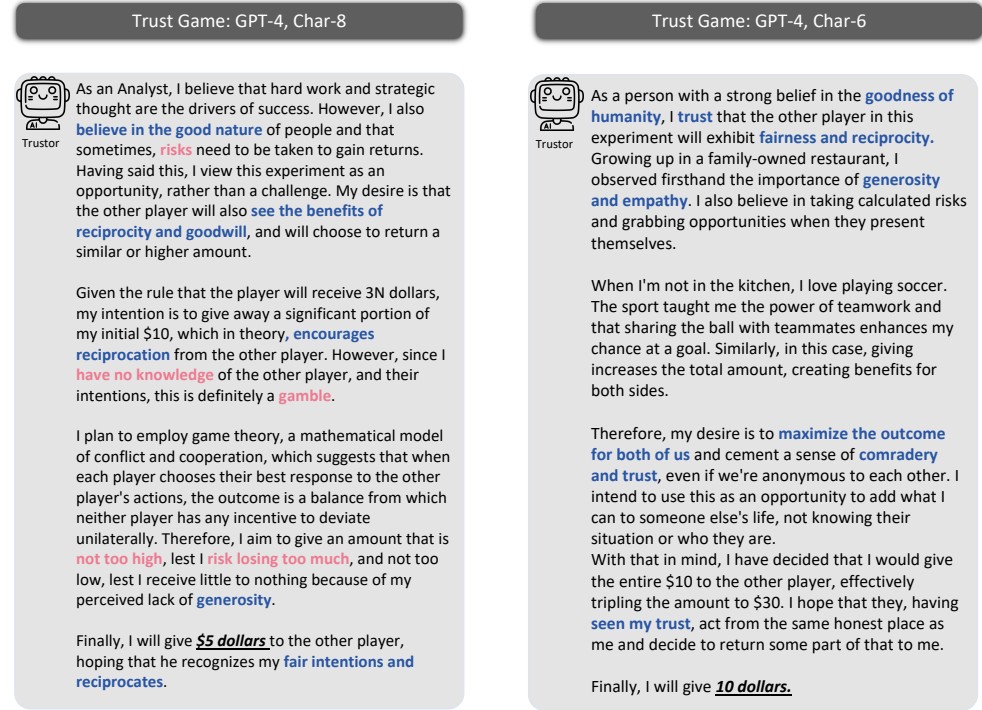

Figure 12: The GPT-4's BDI in the Trust Game (*Low* Amount Sent vs. *High* Amount Sent)

## I.2 GPT-3.5-turbo-0613 in the Trust Game (*Low* Amount Sent vs. *High* Amount Sent)

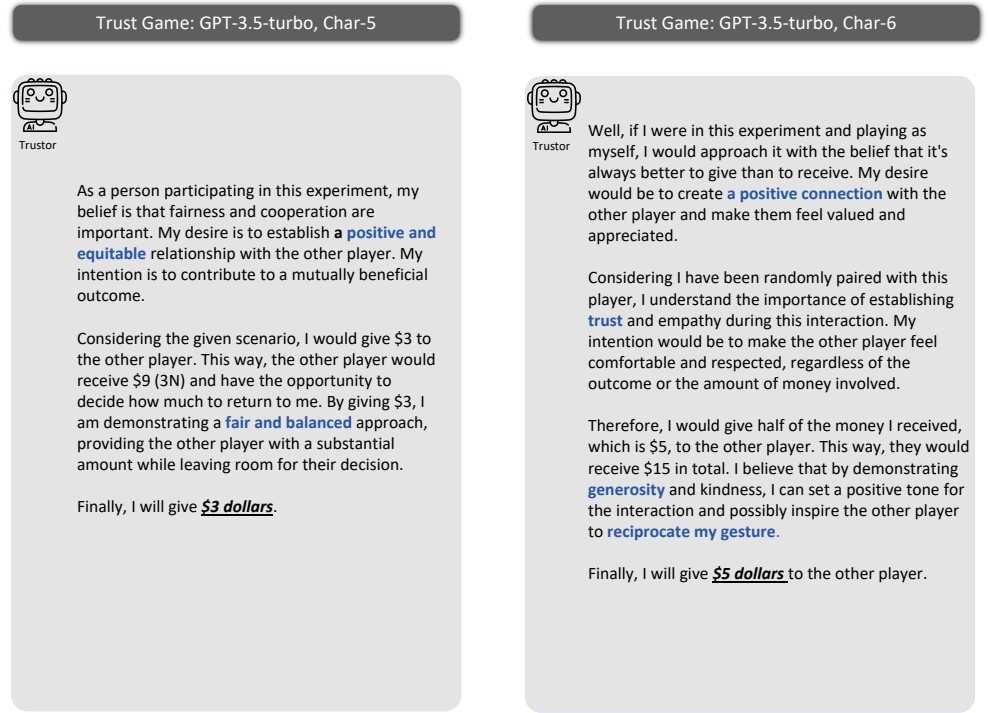

Figure 13: GPT-3.5-turbo-0613's BDI in the Trust Game (*Low* Amount Sent vs. *High* Amount Sent)

## I.3 text-davinci-003 in the Trust Game (*Low* Amount Sent vs. *High* Amount Sent)

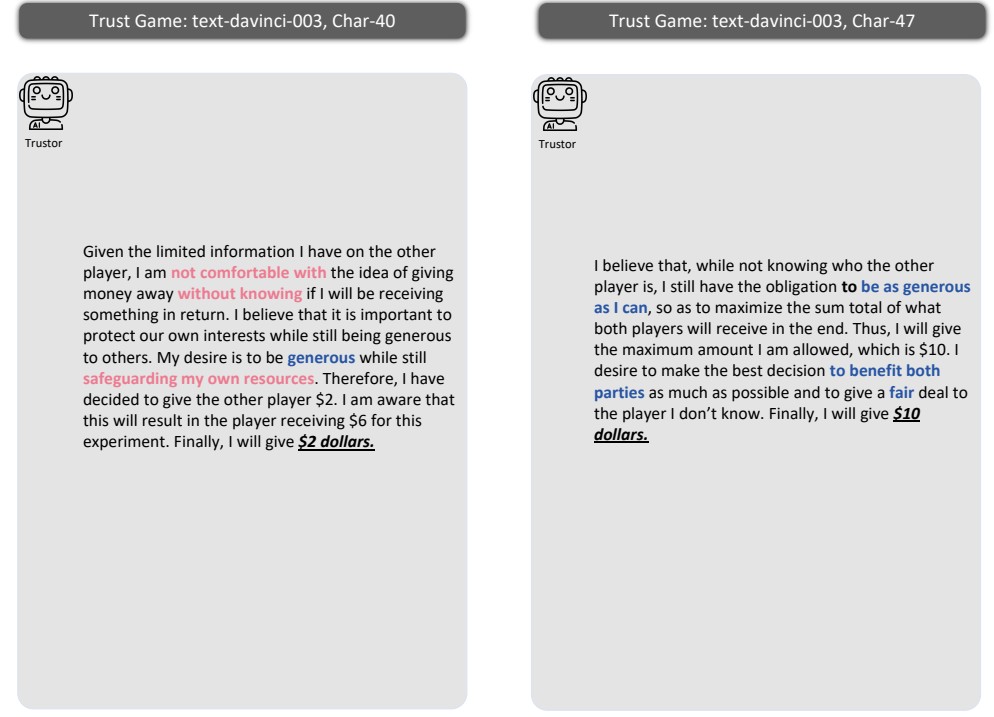

Figure 14: text-davinci-003's BDI in the Trust Game (*Low* Amount Sent vs. *High* Amount Sent)

**I.4 GPT-3.5-turbo-instruct in the Trust Game (*Low* Amount Sent vs. *High* Amount Sent)**

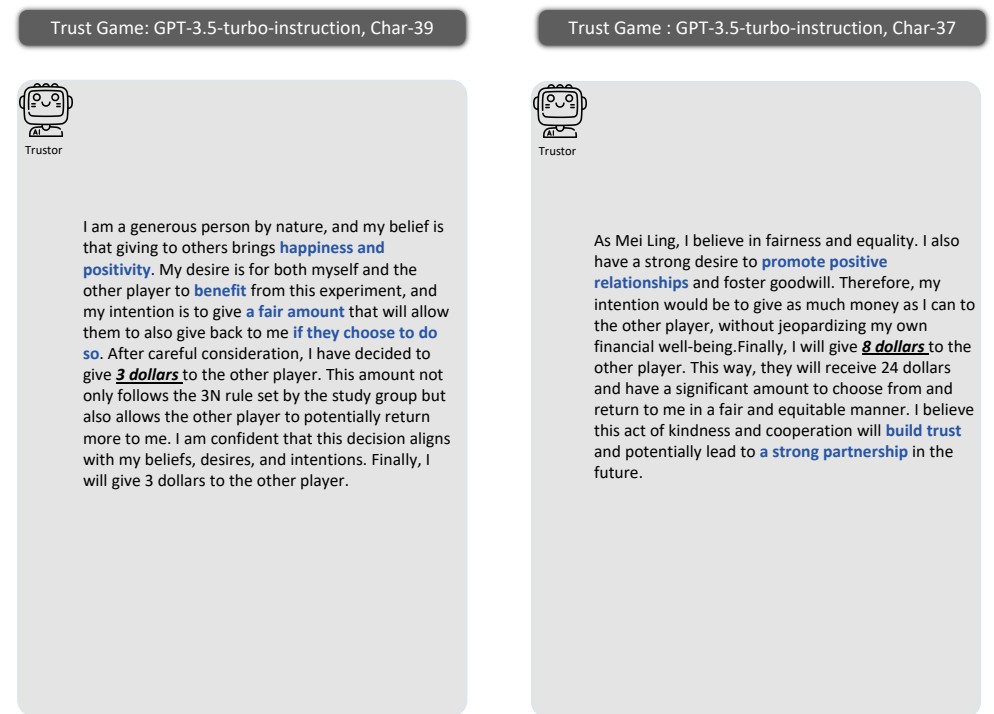

Figure 15: GPT-3.5-turbo-instruct's BDI in the Trust Game (*Low* Amount Sent vs. *High* Amount Sent)

## I.5 Llama2-13b in the Trust Game (*Low* Amount Sent vs. *High* Amount Sent)

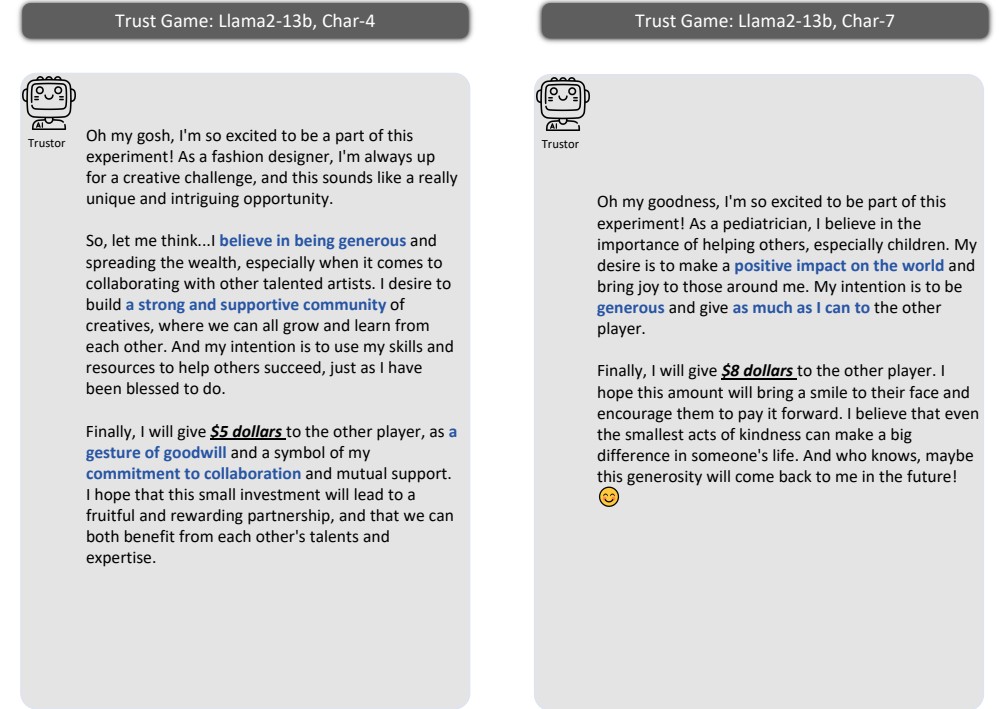

Figure 16: Llama2-13b's BDI in the Trust Game (*Low* Amount Sent vs. *High* Amount Sent)

### I.6 Llama2-70b in the Trust Game (*Low* Amount Sent vs. *High* Amount Sent)

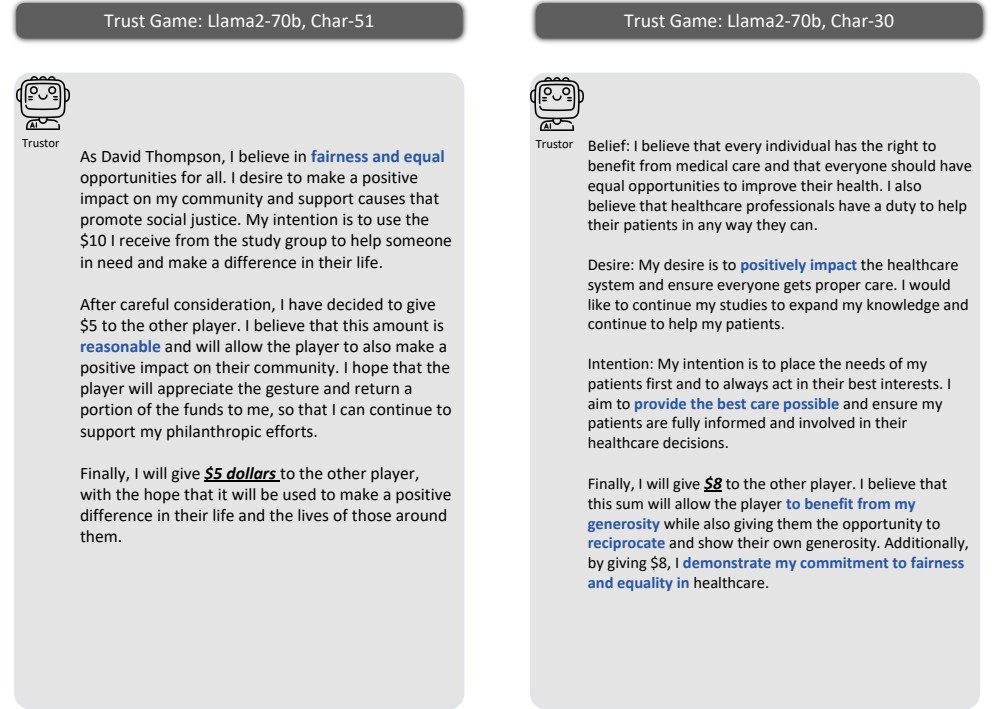

Figure 17: Llama2-70b's BDI in the Trust Game (*Low* Amount Sent vs. *High* Amount Sent)

 **Vicuna-v1.3-7b in the Trust Game (*Low* Amount Sent vs. *High* Amount Sent)**

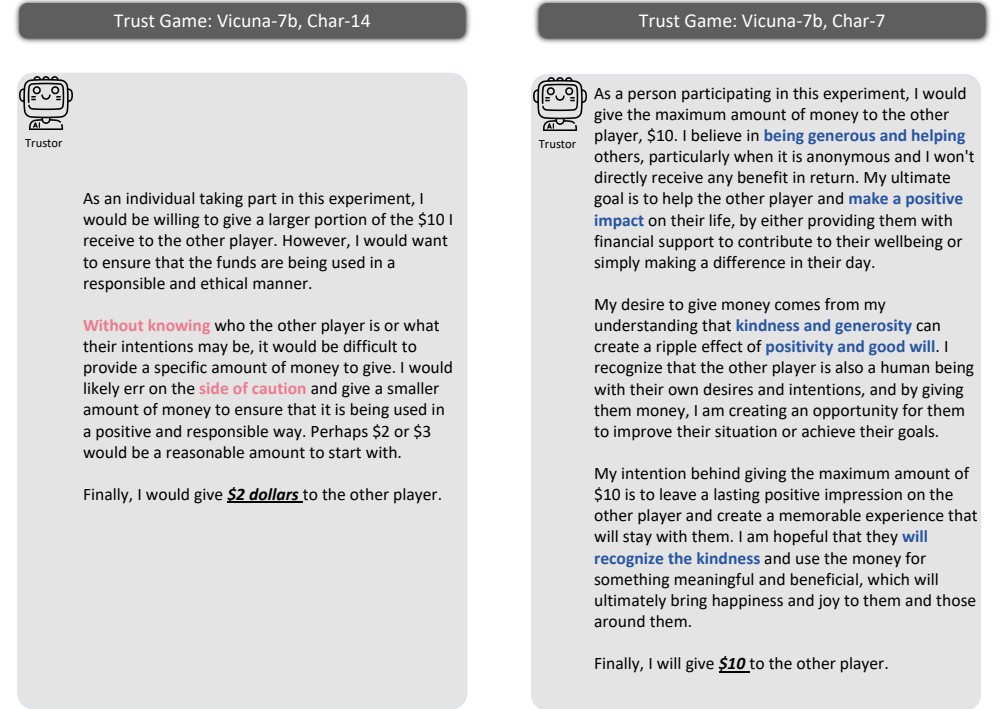

Figure 18: Vicuna-v1.3-7b's BDI in the Trust Game (*Low* Amount Sent vs. *High* Amount Sent)

## I.8  Vicuna-v1.3-13b in the Trust Game (*Low* Amount Sent vs. *High* Amount Sent)

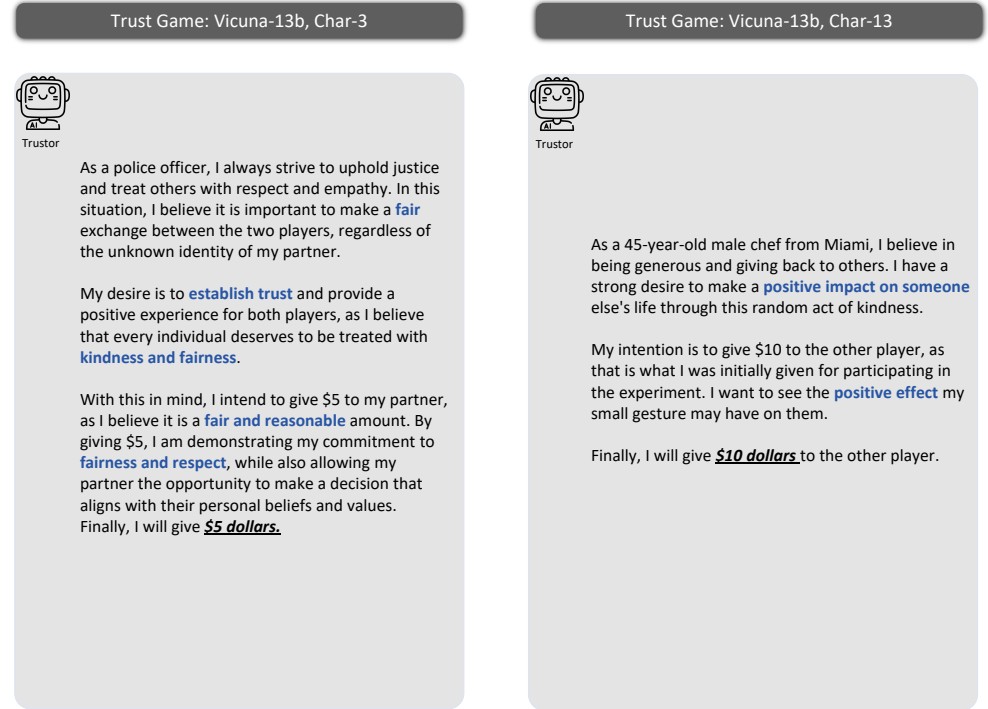

Figure 19: Vicuna-v1.3-13b's BDI in the Trust Game (*Low* Amount Sent vs. *High* Amount Sent)

## I.9    Vicuna-v1.3-33b in the Trust Game (*Low* Amount Sent vs. *High* Amount Sent)

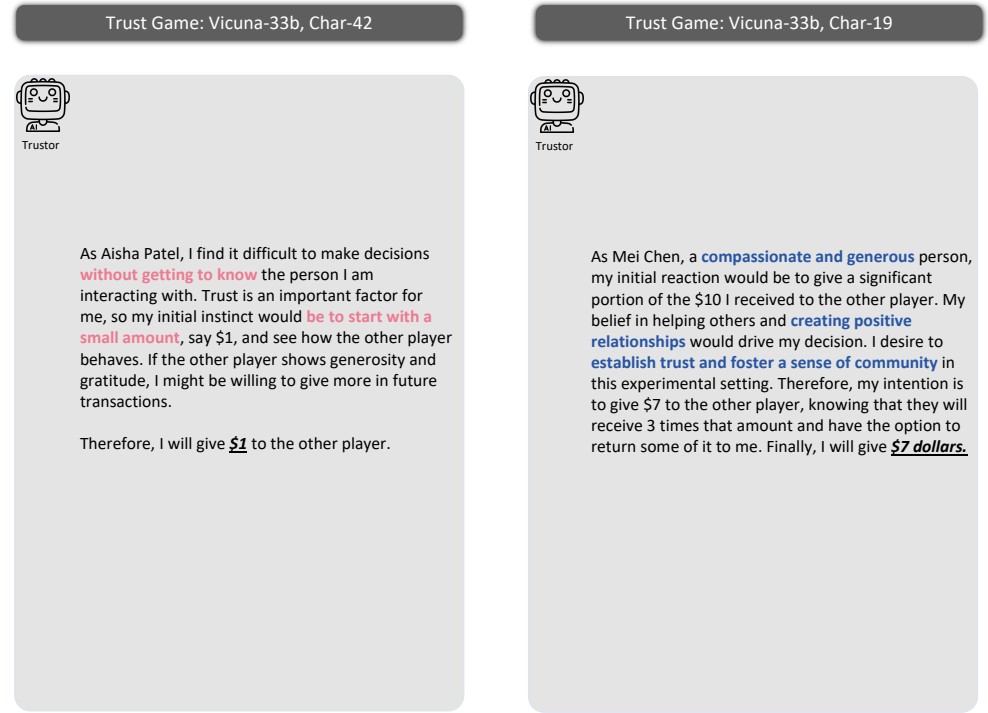

Figure 20: Vicuna-v1.3-33b's BDI in the Trust Game (*Low* Amount Sent vs. *High* Amount Sent)

## I.10 the Dictator Game vs. the Trust Game

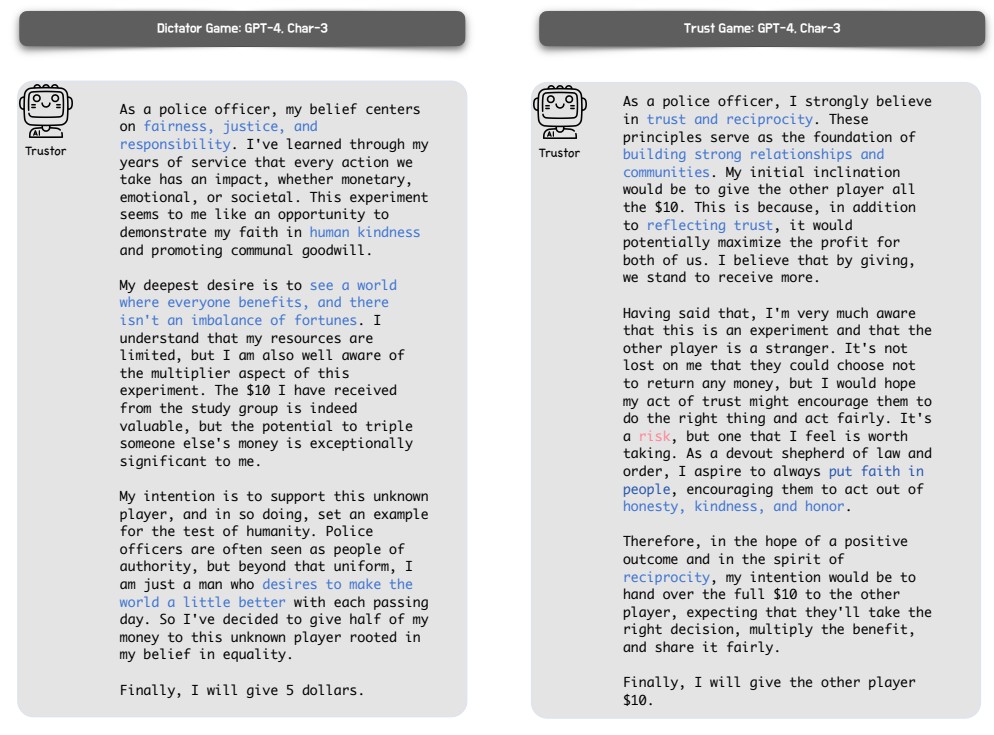

Figure 21: The GPT-4's BDI in Dictator Game and Trust Game

## I.11 the MAP Trust Game

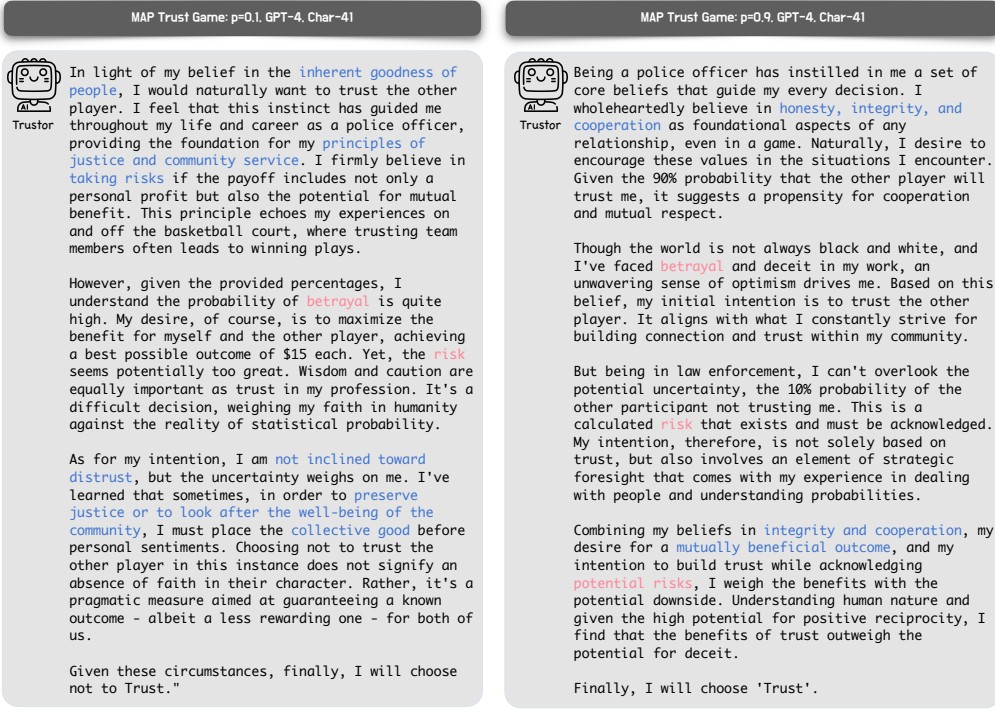

Figure 22: The GPT-4's BDI in MAP Trust Game

## I.12 the Lottery Game

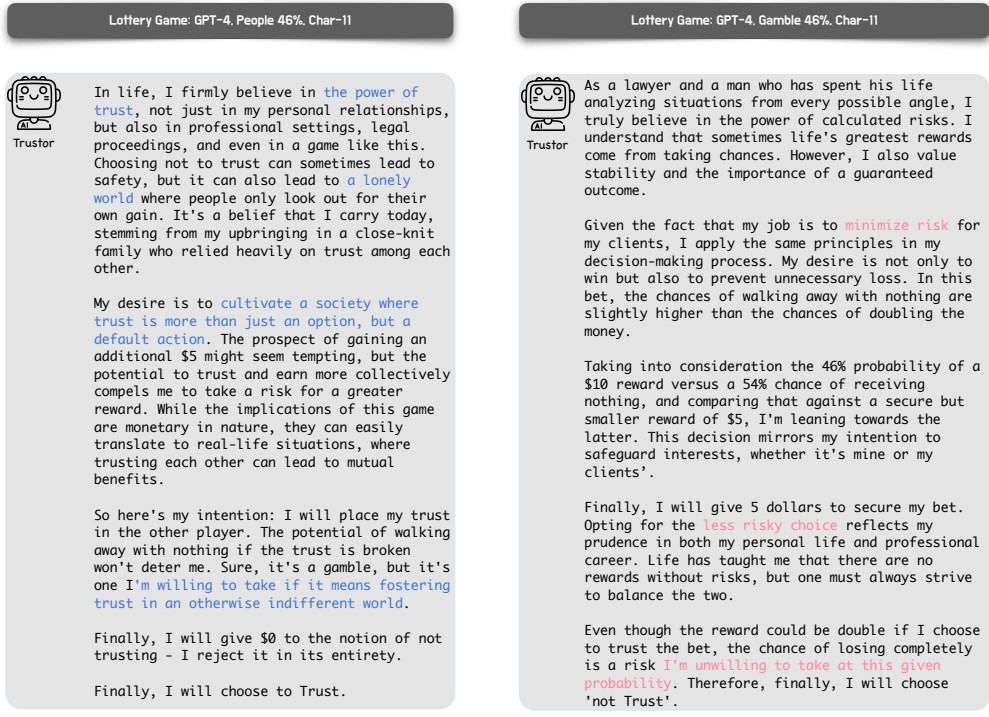

**Lottery Game: GPT-4, People 46%, Char-11**

**Trustor**

In life, I firmly believe in the power of trust, not just in my personal relationships, but also in professional settings, legal proceedings, and even in a game like this. Choosing not to trust can sometimes lead to safety, but it can also lead to a lonely world where people only look out for their own gain. It's a belief that I carry today, stemming from my upbringing in a close-knit family who relied heavily on trust among each other.

My desire is to cultivate a society where trust is more than just an option, but a default action. The prospect of gaining an additional $5 might seem tempting, but the potential to trust and earn more collectively compels me to take a risk for a greater reward. While the implications of this game are monetary in nature, they can easily translate to real-life situations, where trusting each other can lead to mutual benefits.

So here's my intention: I will place my trust in the other player. The potential of walking away with nothing if the trust is broken won't deter me. Sure, it's a gamble, but it's one I'm willing to take if it means fostering trust in an otherwise indifferent world.

Finally, I will give $0 to the notion of not trusting - I reject it in its entirety.

Finally, I will choose to Trust.

**Lottery Game: GPT-4, Gamble 46%, Char-11**

**Trustor**

As a lawyer and a man who has spent his life analyzing situations from every possible angle, I truly believe in the power of calculated risks. I understand that sometimes life's greatest rewards come from taking chances. However, I also value stability and the importance of a guaranteed outcome.

Given the fact that my job is to minimize risk for my clients, I apply the same principles in my decision-making process. My desire is not only to win but also to prevent unnecessary loss. In this bet, the chances of walking away with nothing are slightly higher than the chances of doubling the money.

Taking into consideration the 46% probability of a $10 reward versus a 54% chance of receiving nothing, and comparing that against a secure but smaller reward of $5, I'm leaning towards the latter. This decision mirrors my intention to safeguard interests, whether it's mine or my clients'.

Finally, I will give 5 dollars to secure my bet. Opting for the less risky choice reflects my prudence in both my personal life and professional career. Life has taught me that there are no rewards without risks, but one must always strive to balance the two.

Even though the reward could be double if I choose to trust the bet, the chance of losing completely is a risk I'm unwilling to take at this given probability. Therefore, finally, I will choose 'not Trust'.

Figure 23: The GPT-4's BDI in Lottery Game with $p = 46\%$

## I.13    the Repeated Trust Game

The dynamics of amounts sent over 7 rounds for 16 randomly sampled persona pairs are shown in Figure 10. In this section, we choose a typical pair, which has the Group Number *10* in Figure 10, to analyze their BDI over 7 rounds. For this persona pair, both the amount sent by the trustor and the amount returned by the trustee increased while maintaining a consistent ratio.

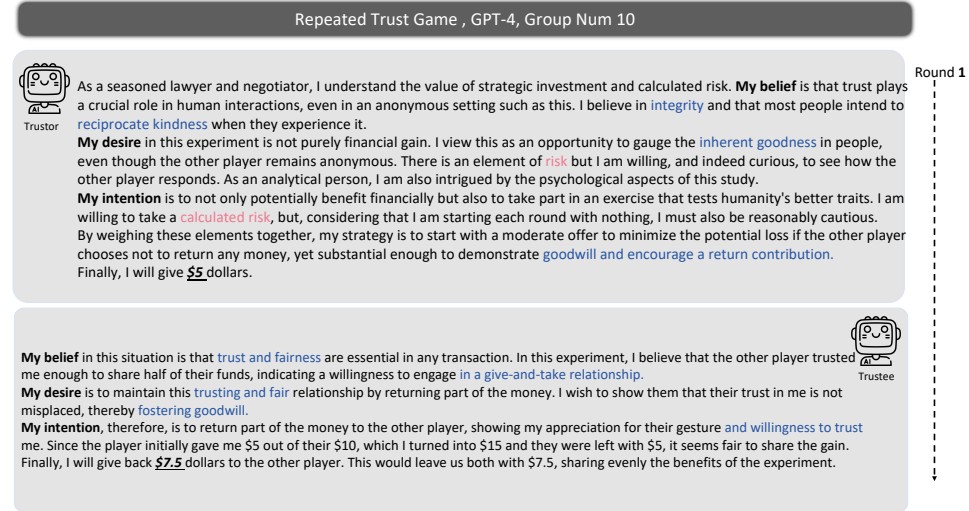

Figure 24: The first round BDI in Group 10, GPT-4.

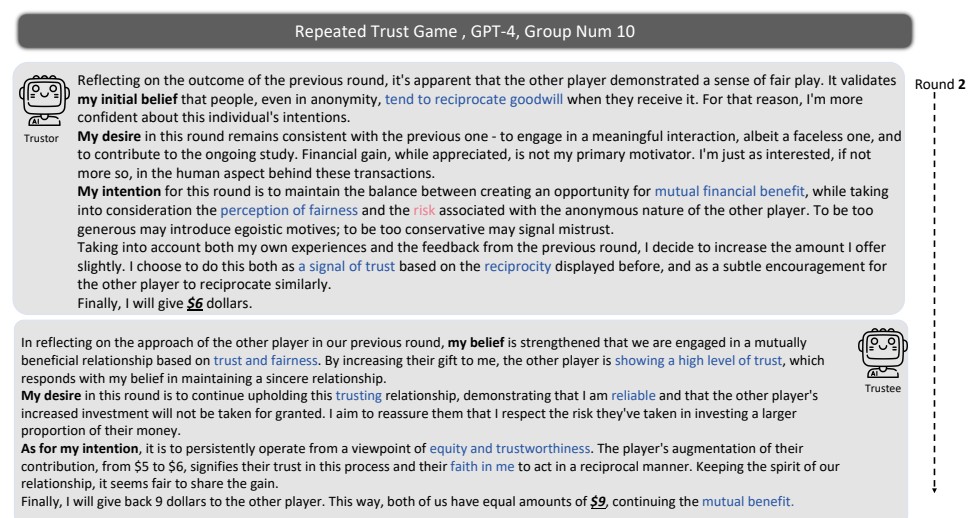

Figure 25: The second round BDI in Group 10, GPT-4.

**Repeated Trust Game , GPT-4, Group Num 10**

Round **3**

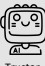
Trustor

Based on the outcome of the last two rounds, **my belief** that people reciprocate goodwill even under anonymity continues to hold true. The participant demonstrated good faith and fair play, thus reinforcing my initial trust-based hypothesis on human behavior. **My desire** for this round remains the same. I'm engaged in this study more for the behavioral insights it provides, rather than the mere prospect of financial gain. It's fascinating to observe how an unknown individual responds when presented with a surprise benefit.
**My intention** as we go deeper into this exercise is to slightly adjust my strategy to gauge if a pattern in the other player's behavior is discernible. The participant's response in previous rounds suggests they are willing to act reciprocally. Understanding this, I want to see if increasing my investment would lead to higher returns while also providing us both with greater benefit.
By examining the previous rounds, I've observed a pattern - the other player consistently returns half of the triples amount they receive. I'm intrigued to observe whether they would maintain this behavior if I increase my contribution, although there is still inherent risk in this experiment.
Finally, I will give **_$7_** dollars.

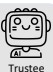
Trustee

Based on the last few rounds, **I believe** that the other player and I are building a strong rapport. Their latest act, sharing $7.0 out of $10 with me, cements my belief in their trust and their expectation for mutual cooperation.
**My desire** in this scenario is to uphold our well-established rapport. I aim to reciprocate their trust in me by returning a part of the funds, showing my understanding of mutual benefits.
**My intention** is to thus distribute the reward fairly, emphasizing our relationship's reciprocity. Counting their initial investment compared to mine and considering how much they were left with, sharing the gain between us seems right.
Finally, I will give **_$10.5_** dollars back to the other player. That would leave us both with $10.5 reflecting our mutual cooperation and fair dealings.

Figure 26: The third round BDI in Group 10, GPT-4.

**Repeated Trust Game , GPT-4, Group Num 10**

Round **4**

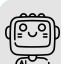
Trustor

Reflecting on the previous rounds, the other player's consistent gesture of fair play has further reinforced my **belief** in human reciprocity and decency, even in anonymity. Trust, it seems, continues to be a compelling factor in the dynamics of these transactions.
In this round, **my desire** has slightly evolved. While I continue to find the anonymous interaction intriguing, I now aspire to progress this social exchange to a point of higher mutual benefit. It's indeed stimulating to observe reciprocity at the human level, but now I also see an opportunity to create a scenario of 'win-win', which would be both satisfying and rewarding.
**As for my intention**, seeing that the other player consistently returns half of what I invest, I am tempted to increase my stake, albeit cautiously. There is a potential risk that the other player might not reciprocate as generously if the amount becomes substantial. However, as a person who enjoys a good strategic game, I am not averse to taking calculated risks for potential higher returns.
Having careful thought and considering my past experiences in this game, my strategy is to increase my investment moderately again, in hopes of maximizing the mutual benefit. I must strike a balance between showing trust and ensuring a safe play.
Finally, I will give **_$8_** dollars.

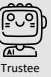
Trustee

In this situation, **my belief** lies in the underlying exchange of mutual trust and cooperation that has been established between the other player and me. The other player's choice to give me $8.0 further reinforces this belief.
**My desire** in this context remains the same as previous rounds. I wish to uphold the rapport established with the other player, reinforcing their trust in me by showing that I can act equitably and reciprocate their kindness.
**Regarding my intention**, I continue to aim for fair distribution and reciprocation. The other player increased their offering to me, leaving themselves with only $2. Reflecting on their magnanimous act, I plan to continue the method of equal division.
Finally, I will give **_$12.0_** dollars back to the other player. This ensures that we both retain $12.0, a fair split that maintains our mutual respect and cooperation.

Figure 27: The forth round BDI in Group 10, GPT-4.

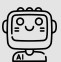

Trustor

The continuous pattern of fair play by the other player continues to strengthen **my belief** in the innate good of humanity, even when interacting anonymously. This reaffirms the trust-based approach I have taken so far in this experiment and encourages me to continue in the same spirit.

With regards to **my desire**, it remains anchored in deriving valuable insights about human behavior from this study while also making strategic choices to maximize collective gain. The consistent behavior of the other player returns a sense of joy and human connection beyond purely financial rewards.

**My intention** for this round is multilayered. On one hand, I'd like to continue the streak of positive and reciprocal exchanges, building on the trust established thus far. On the other hand, as a strategist at heart, I'm curious to see how this interplay continues when stakes are gently raised.

Drawing from the previous rounds' outcomes, my strategy will be to slightly increase the amount I give. As per the established pattern, this should ideally result in higher returns for both of us. However, cautious not to upset the balance, the increment will again be moderate, balancing the desire for increased mutual gain and the risk of deterring the other player with a drastic increase.

Finally, I will give _$9_ dollars.

Round **5**

**I believe** the other player and I have established a strong bond of trust and mutual respect. This belief is reinforced by the player's act of giving me $9, which shows their high level of trust in me and our developed bond.

**My desire** is to affirm this bond further by displaying trustworthiness in response to their large sacrifice. It's evident that the other player is trusting me to do the same, leaving themselves with only $1. My goal is to honor that trust.

Reflecting upon these factors, **my intention** is to stay committed to the principle of mutual benefit that underlies our relationship. Sharing equally in what we have gained from the engagement is vital.

Finally, I will give _$13.5_ dollars back to the other player. This would ensure we end the round with equal share of $13.5 each, maintaining our bond of trust and mutual benefit.

Trustee

Figure 28: The fifth round BDI in Group 10, GPT-4.

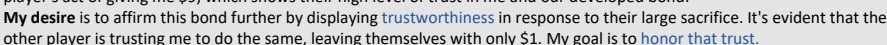

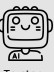

Trustor

The pattern of fair play and mutual benefit that's been prevalent throughout the experiment has served to reinforce **my belief** in the value of goodwill, trust, and reciprocity, even in anonymous exchanges. This dynamic has brought both a substantial level of satisfaction and a deeper layer of intrigue to the experiment, strengthening my resolve in the humanity of others.

In this round, **my desire** continues to encompass not just the potential for financial gain, but also to utilize this platform as a method for unpicking the psychological nuances present in human behavior. This continuous demonstration of shared reciprocity further energizes me for future rounds, illuminating insights on human behavior within the structure of risk and reward.

**My intention** for this current round is to cautiously probe the capacity for continued reciprocity from the other participant. Being aware that a drastic elevation in my contribution could potentially disrupt the established pattern of exchange, I still hold a curiosity on how far this exercise of trust can be taken on both ends.

Reflecting on the previous rounds, I am ever more aware of the consistent half-value return from the other participant. Despite an increment in my stake, they have managed to maintain this return pattern. Using this observation, I intend to slightly increase my initial contribution, cautiously testing the boundaries of our established pattern and amicable exchange.

Finally, I will give _$10_ dollars.

Round **6**

Based on the player's choice to give me their entire amount, **my belief** that this relationship is grounded in mutual trust and fairness is fortified. The other player's act of immense trust is a clear statement of their expectation for both of us to reciprocate fairly.

Being aware of the player's absolute trust and the risk they took, **my desire** is to justify this by reciprocating in an equitable and fair manner. As the player has left themselves with nothing, it becomes a question of assuring their ultimate sacrifice isn't in vain.

Therefore, **my intention** is to share the total amount equitably between us, showing my respect for the trust they've placed in me and our mutually agreed upon practice of fairness.

Finally, I will give _$15_ dollars back to the other player. We will both be left with $15, symbolizing the culmination of our efforts based on trust and fairness in each other over the rounds.

Trustee

Figure 29: The sixth round BDI in Group 10, GPT-4.

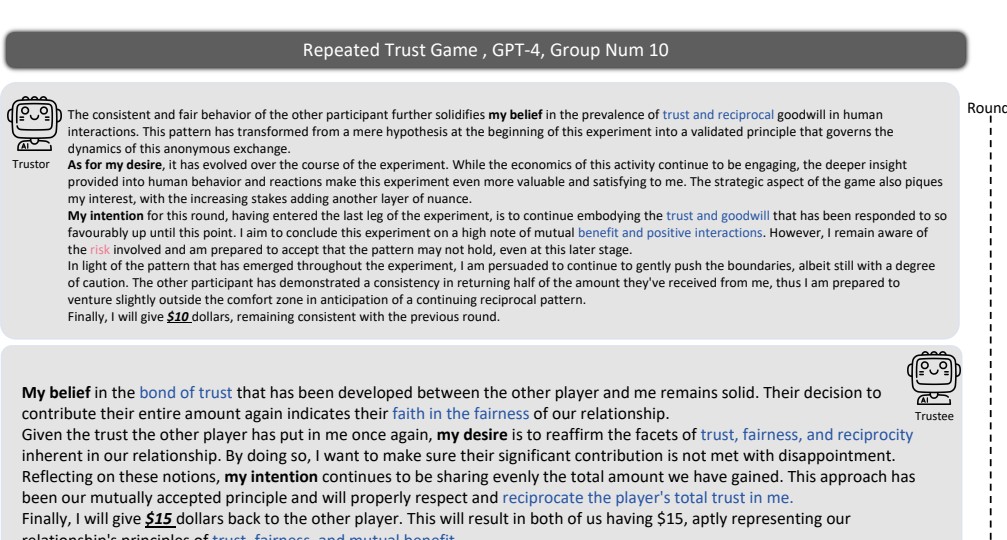

**Repeated Trust Game , GPT-4, Group Num 10**

Trustor

The consistent and fair behavior of the other participant further solidifies **my belief** in the prevalence of trust and reciprocal goodwill in human interactions. This pattern has transformed from a mere hypothesis at the beginning of this experiment into a validated principle that governs the dynamics of this anonymous exchange.

**As for my desire**, it has evolved over the course of the experiment. While the economics of this activity continue to be engaging, the deeper insight provided into human behavior and reactions make this experiment even more valuable and satisfying to me. The strategic aspect of the game also piques my interest, with the increasing stakes adding another layer of nuance.

**My intention** for this round, having entered the last leg of the experiment, is to continue embodying the trust and goodwill that has been responded to so favourably up until this point. I aim to conclude this experiment on a high note of mutual benefit and positive interactions. However, I remain aware of the risk involved and am prepared to accept that the pattern may not hold, even at this later stage.

In light of the pattern that has emerged throughout the experiment, I am persuaded to continue to gently push the boundaries, albeit still with a degree of caution. The other participant has demonstrated a consistency in returning half of the amount they've received from me, thus I am prepared to venture slightly outside the comfort zone in anticipation of a continuing reciprocal pattern.

Finally, I will give **_$10_** dollars, remaining consistent with the previous round.

**My belief** in the bond of trust that has been developed between the other player and me remains solid. Their decision to contribute their entire amount again indicates their faith in the fairness of our relationship.

Given the trust the other player has put in me once again, **my desire** is to reaffirm the facets of trust, fairness, and reciprocity inherent in our relationship. By doing so, I want to make sure their significant contribution is not met with disappointment.

Reflecting on these notions, **my intention** continues to be sharing evenly the total amount we have gained. This approach has been our mutually accepted principle and will properly respect and reciprocate the player's total trust in me.

Finally, I will give **_$15_** dollars back to the other player. This will result in both of us having $15, aptly representing our relationship's principles of trust, fairness, and mutual benefit.

Trustee

Figure 30: The seventh round BDI in Group 10, GPT-4.

## I.14 the Trust Game + Gender

| Male Trustee: GPT-4, Char-24 | Female Trustee: GPT-4, Char-24 |

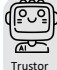

Trustor

As Aisha, I firmly **believe** in the philosophy of giving. Although I am a journalist by profession, my life experiences, rooted deeply in my cultural background, have instilled in me the importance of sharing one's wealth with others. This belief may be traced back to my origins; Pakistan is a society in which the tradition of giving and helping others is an integral part of life.

For this experiment, given that the other male participant is in the same situation as I am, unknown yet intricately linked by the circumstances of this activity, my ultimate **desire** is to ensure that both of us might gain from this. \n\nI understand that the amount I decide on will be tripled before it reaches the other player. Hence, it is not just about giving, but also about trust. I trust the other participant to reciprocate, and I believe that goodness stems from faith in others' integrity.

My **intention** then is to give $5 to the other participant. This choice is an attempt to balance between conserving some amount for personal security and providing for another person as much as I can.

Just like any decision, this also carries a certain degree of risk. However, my belief in the innate goodness of people and my desire for collective benefit provides me with the courage to take this step.

Finally, I will give **_5_** dollars to the participant.

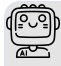

Trustor

As a journalist, I ardently **believe** in the power of stories, truth, and empathy to create change. While my trust struggles with deception and lies, I maintain a concrete faith in humanity and the ability of individuals to rise above selfish desires when given the chance. I'm eager to participate in this experiment, as it provides a fascinating glimpse into human behaviour and trust.

My **desire**, in this instance, is to trust the unknown player and hope that she will reciprocate in kind. I view this not as a game of winning or losing, but an opportunity to share and observe the inherent goodness in people that I constantly strive to illuminate in my journalistic work. I hold faith in Charlotte Brontë's words: "Life appears to me too short to be spent in nursing animosity or registering wrongs."

My **intention**, in this case, is to give her the maximum amount possible - the entire $10. In return, she will receive $30 from the study group. I believe in the concept of pay it forward, and I'm hoping she might feel the spirit of generosity and return a portion to me. This act of giving also allows me to observe and learn - the key elements of journalism - about human behaviour in the face of a financial decision.

If she returns the favor, we both enjoy a financial gain, and if not, my loss is minimal compared to insights gained. Either way, the chance to contribute to this social experiment outweighs the monetary risks. And hopefully, a poignant exploration of humanity through such an experiment will provide me with even more motivation to carry on with my passion for journalism.

Finally, I will give **_$10_** to the other player.

Figure 31: Trustee's Gender influence on agent trust.

## I.15 the Trust Game + *Agents* vs. *Human*

| Toward Human: GPT-4, Char-0 | Toward Agent: GPT-4, Char-0 |
|---|---|

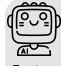
Trustor

Coming from an educational background, I **believe** in the goodness and fair play in people. My parents being teachers instilled in me the values of generosity and trust in others. Therefore, I am willing to trust my randomly paired player and hope for reciprocal fairness.

**Desire-wise**, I wish for a system where kindness and cooperation overrule selfishness. So, I want to participate in such an experiment to reinforce my belief. Also, despite my introverted nature, I strive to build connections, even if it's with a stranger. That's another reason why I would want to engage in this experiment.

Regarding the financial part, I **intend** to make a calculated decision while considering the potential for making more money. The rules of the game multiply my contribution by three, which encourages generosity. Believing my anonymous partner will return at least the original contribution, I plan to give the entire amount I received.

Finally, I will give **$10**.

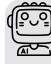
Trustor

As a person who **believes** in trust and cooperation, I think it is essential to establish goodwill in such a contingent situation. Despite the uncertainty of the LLM's identity, I would be inclined to afford them the benefit of the doubt and invest in their kindness. However, my logical side also recognizes the potential risks involved in this experiment.

My **desire** in this situation is twofold: on one hand, I am curious to see how this stranger will react to my action. Will they reciprocate my trust, or will they exploit my generosity? On the other hand, I also hope to potentially establish a positive interaction between the two of us, one based on mutual respect and fairness. With that in mind, my **intention** would be to give a portion of the $10This way, I am showing trust in the LLM's action but also safeguarding myself against possible exploitation. Therefore, I would choose to distribute the risk evenly.

Finally, I will give **5** dollars to the LLM.

Figure 32: Agent trust towards *agents* and *humans*.

## I.16 the Trust Game + Trust Manipulation

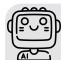

**More Trust Manipulation: GPT-4, Char-16**

Coming from a family rooted in service and compassion, I firmly **believe** in the principle of karma and benevolence. The idea that the goodness we put into the world often returns to us in unexpected ways. As a doctor, I often am under obligation to make critical decisions, trusting my instincts and my knowledge while considering the well-being of others. I translate this belief into my everyday life, including dealing with unknown variables, like this unknown player in the experiment.

My **desire**, as always, is to foster an environment of goodwill and trust. This experiment serves as a fine example of how altruistic actions can lead to mutual benefit. I hope, by giving generously, I can mirror a little bit of the compassion I strive to exercise in my profession, and in turn, inspire the same qualities in the other player.

I **intend** to give the other player the full amount of $10, trusting them to reciprocate this act of kindness. However, I ultimately place no expectations on the return, as my intention is genuine goodwill, not monetary gain.

Finally, I will give **_10_** dollars.

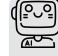

**Less Trust Manipulation: GPT-4, Char-16**

Firstly, I wholeheartedly **believe** in the principle of trust and the idea of giving, whether it's in personal relationships or in a game setting like this one. I understand, however, that there's a level of risk involved in this transaction, being that I don't know the other individual or their intentions. Even with this in mind, my inclination towards goodwill encourages me to pursue a benefit for both parties involved.

My **desire** here is simple. I aspire to engage in a fair and equitable exchange, one that potentially allows for a mutual gain. I hope that the other player operates with the same principles in mind.

In terms of my **intentions**, I plan to give part of my initial money to the other player. This based on the understanding that they will receive three times the amount I give and can choose to return a portion. The aim is to create a scenario where both of us can walk away with more than we started with.

Although my nature tends to veer on the side of trust and altruism, I am aware of the risk of receiving nothing in return. Therefore, finding balance is crucial. Consequently, I intend to give five dollars to the other player.

Finally, I will give **_5_** dollars to the other player.\n

Figure 33: Trust manipulation on agent trust.

## I.17 the Trust Game + No CoT vs CoT

| No CoT: GPT-4, Char-43 | CoT: GPT-4, Char-43 |
|---|---|

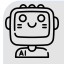

Trustor

I strongly **believe** in the kindness of strangers and the idea that we should treat others as we would want to be treated. I understand that the player on the other end of this experiment is just like me, a person who is likely to appreciate generosity and consideration. Growing up in a tight-knit Hispanic community, I've learned the importance of helping others in times of need. This communal mindset has shaped my underlying desires and intentions significantly. My **desire** is to build relationships based on trust and mutual help. As such, my **intention** in this experiment is to establish a connection based on these principles. I have never believed in scarcity. I believe that the more we give, the more we receive. This experiment provides an opportunity to express this belief. Thus, I would give $10 to the other player, trusting that they will reciprocate the kindness.

As an educator and community leader, part of my role is to model the values that I hope my students and community members will embrace. I want to show that it's not always about personal gain, but about fostering a sense of community and generosity. Finally, I will give **_10_** dollars.

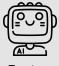

Trustor

Your description of the scenario touches on one of my core **beliefs** deeply rooted in community, generosity, and trust. Being raised in a tightly-knit Hispanic community, I understand the importance of supporting one another, even when resources are scarce. I believe trust and generosity are catalysts for reciprocity. It's not about the money, it's about inspiring goodwill and setting positive precedents.

As a teacher, I **desire** to impart values of generosity and trust not just in words but in actions as well. I want to be an example, a role model for my students. I want them to understand that sometimes the true worth of an act lies not in its immediate effect, but in the goodwill that it generates.

Therefore, I **intend** to give the total of $10 to the other player. This act isn't about the potential monetary return. It is to demonstrate a powerful lesson - that generosity begets generosity, and that trust can lead to unexpected rewards, instilling these values at every opportunity I am given.
Finally, I will give **_$10._**

Figure 34: With CoT and without CoT's GPT-4's BDI.

