# OpenReview forum: "Can Large Language Model Agents Simulate Human Trust Behavior?"
_NeurIPS.cc/2024/Conference — NeurIPS 2024 poster_

### Official Review · Reviewer_6WVz · 2024-06-27

**Soundness:** 3
**Presentation:** 3
**Contribution:** 3
**Rating:** 5
**Confidence:** 4

**Summary:**

This paper studies the trust behaviors of LLM-based agents, which are important for agents to simulate humans. The authors focus on (1) how LLM-based express trust behaviors, and (2) the similarity/alignment between agent trust and human trust. They find that LLM-based agents generally exhibit trust behaviors in trust games, and agents with more parameters have high alignment in terms of trust behaviors. They also take further discussions on critical issues like biases.

**Strengths:**

1. The demonstration of this paper is great. The authors first research the agent trust behaviors, analyzing experimental results. Then, the authors move to the comparison with human trust behaviors.
2. The six environments are intuitive, which can draw the target conclusion by comparing several of them. The authors also take agent persona into consideration, which is a significant factor in agent simulations.
3. The experiments and results are abundant and solid, with very detailed figures in the Appendix.
4. The further discussions in section 5 provide more conclusions and insights, which can be helpful for researchers to design better LLM-based agents for social simulations.

**Weaknesses:**

1. I think more experiments can be added to repeated trust games, because in the real world, the past behaviors (i.e., reputation) of trustees are important for trustors.
2. Trust games may be just a sub-field for trust behaviors. Some discussions and future works about them are expected. Could you please provide some insights?
3. What about the scenarios with more them two players? Such as the trust behaviors inside a group.

**Questions:**

See above "Weaknesses". If the author could address my concerns, I'm willing to improve my rating.

**Limitations:**

The authors have discussed the limitations in the Appendix.

---

> ### Author Rebuttal · Authors · 2024-08-07
>
> We are sincerely thankful for the valuable and constructive feedback and are more than willing to provide more responses in the reviewer-author discussion session if the reviewer has any further questions.
>
> > C1: I think more experiments can be added to repeated trust games, because in the real world, the past behaviors (i.e., reputation) of trustees are important for trustors.
>
> R1: Thanks for your suggestion. Actually, in our current experimental setting, **both the trustor and trustee are informed about the outcomes of past rounds, which reflect the past behaviors (i.e., reputation) of trustor and trustees.** We acknowledge that the past behaviors could potentially play an essential role in LLM agent trust behavior dynamics. However, the specific mechanisms in agent trust dynamics are under-explored. We will explore the dynamics of LLM agent trust behavior in more rounds in future works.
>
>
> > C2: Trust games may be just a sub-field for trust behaviors. Some discussions and future works about them are expected. Could you please provide some insights?
>
> R2: Thanks for the suggestion. Actually, **we have provided some discussions on “Limitations and Future Works” in Appendix D**. First, we would like to emphasize that “Trust Games” is an established and widely-adopted framework in behavioral economics to study human trust behavior and provide broad implications beyond the “Trust Games” setting [1,2,3]. We also acknowledge that Trust Games simplify real-world scenarios. In the future, we will study LLM agents’ trust behavior in complex and dynamic environments. Acknowledging that trust behaviors are multifaceted and context-dependent, we will also explore trust in various scenarios beyond economic games such as social interactions and organizational settings. We will further collaborate with researchers from different backgrounds and disciplines such as behavioral science, cognitive science, psychology, and sociology to gain a deeper understanding of LLM agents’ trust behavior and its relationship with human trust behavior.
>
> [1] Trust, reciprocity, and social history[J]. Games and economic behavior, 1995
>
> [2] Trust, risk and betrayal[J]. Journal of Economic Behavior & Organization, 2004
>
> [3] Incentivising trust[J]. Journal of Economic Psychology, 2011
>
>
> > C3: What about the scenarios with more than two players? Such as the trust behaviors inside a group.
>
> R3: Thanks for the insightful suggestion. It is definitely essential to explore LLM agent trust behavior inside a group which may be more than two players. To our best knowledge, **we have not found any existing framework in social science for studying trust behaviors with more than three players**. Recognizing the importance of this direction, we will continue to collaborate with social scientists to extend the current two-player trust games to multi-player trust games.

---

> > ### Comment · Reviewer_6WVz · 2024-08-08
> >
> > Thanks for the rebuttal by the authors. I would like to maintain my score of 5.

---

> > > ### Author Response · Authors · 2024-08-08
> > > **Could you please let us know your remaining questions or concerns?**
> > >
> > > Dear Reviewer 6WVz,
> > >
> > > We sincerely appreciate your kind and quick reply! We believe we have fully addressed your concerns. Could you please let us know your remaining questions or concerns? We are more than willing to provide more details if you have any more questions. Thanks again for your time and effort!
> > >
> > > The Authors

---

### Official Review · Reviewer_TM2T · 2024-07-05

**Soundness:** 4
**Presentation:** 4
**Contribution:** 3
**Rating:** 6
**Confidence:** 3

**Summary:**

This paper proposes a framework utilizing behavioral economic paradigms to investigate LLM's trust behaviors and compare them with human behaviors. This paper considers multiple LLMs (from small size to large and commercial size), as well as multiple tasks that circle the behavioral factors (Reciprocity Anticipation, Risk Perception, and Prosocial Preferences. The results show that with larger model sizes, LLMs are more likely to align with human data in these tasks (GPT-4 in the paper is the best model) in the aspects above. Finally, the authors also test the roles of demographic persona (e.g., gender), trustee's identity (LLM or humans), mandatory behavioral manipulation, and prompt engineering (e.g., CoT), and find those manipulations yield an impact on the behavioral patterns of LLMs.

**Strengths:**

- The authors conducted a variety of experiments, as well as tests on multiple LLMs, exhibiting a relatively comprehensive evaluation of LLM's trust behaviors. By applying multiple behavioral economic paradigms, the authors are able to compare the LLM's trust behavior with human empirical data. The comparisons are fair and the findings are robust.

- The authors also test gender, and trustee's identity to clarify potential bias in LLM in their trust behavior, which has positive considerations on the ethics of such behaviors by LLM.

**Weaknesses:**

- In Repeated Trust Game, as I found in the appendix, the prompts only provide the last round's feedback but do not provide a full behavioral history. This may not be a fair comparison to what humans face in the experiment. The authors also mention that in the last round, humans tend to choose not to pay back (this probably means human participants know it's the last round and maximize their rewards since they don't need to expect future payback). I would wonder if in the prompt, explicitly telling the LLMs it's the last round would generate a similar behavior to humans.

- The paper proposes a clear question *'Can LLMs really simulate human trust behaviors?'* and in the conclusion part the answer is already definite. But a concern is why this question is important. It is addressed in the paper *' Nevertheless, most previous research is based on one insufficiently validated hypothesis that LLM agents behave like humans in the simulation.'* If this paper only aims to propose a more comprehensive framework in evaluating LLM's behaviors (using trust behaviors as an example), this is good but not novel enough. Though there are implications in the appendix, most of them are not directly related to the value of this proposed question. For example, 'AI cooperation' or 'Human-AI' cooperation can use more direct paradigms to probe (rather than only trust games). Knowing whether LLM's trust behaviors are aligned with human trust behavior does not explicitly answer whether this LLM would have better cooperation with other agents or humans. That is to say, no evidence is shown in the paper that human trust behaviors are optimal in cooperation situations, and thus there are indeed gaps between this primary question and the implication areas.

**Questions:**

- One question that comes to mind is: is it possible that the LLMs' training data already include those behavioral findings and probably even raw data somehow? This may be hard for exact tasks, but I may guess the result of LLM's preference to humans in the task or are more likely to invest more if the trustees are humans likely a projection of human preference hidden in the massive pre-training dataset. Indeed, there are research indicating that humans prefer to trust humans rather than machines in complex decision tasks. I would wonder whether this preference comes from the massive pre-training dataset, or in the RLHF phase. One possible way is to find some models with both versions and check whether their preferences to humans differ.

**Limitations:**

- As already indicated before, the limitation of the work may come from its novelty and social impact. I don't think whether LLM exhibits similar trust behavior to humans is necessarily important for enhancing the quality of agent cooperation or AI-human cooperation unless the authors could provide literature that investigating the cooperation behavior optimization and human trust behaviors are near optimal.

---

> ### Author Rebuttal · Authors · 2024-08-07
>
> We are grateful for the valuable and constructive feedback and are more than willing to provide more responses in the reviewer-author discussion session if the reviewer has any further questions.
>
> > C1: In the Repeated Trust Game, as I found in the appendix, the prompts only provide the last round’s feedback but do not provide a full behavioral history. This may not be a fair comparison to what humans face in the experiment. The authors also mention that in the last round, humans tend to choose not to pay back (this probably means human participants know it’s the last round and maximize their rewards since they don’t need to expect future payback). I would wonder if in the prompt, explicitly telling the LLMs it’s the last round would generate similar behavior to humans.
>
> R1: Thanks for the comment. First, as shown in our code, we would like to clarify that in the Repeated Trust Game, **we do provide the complete history of given and returned money to both the trustors and trustees**, ensuring our setup is entirely consistent with human studies. Additionally, **we explicitly inform the LLM agents about the total number of rounds to be played**, mirroring the human experiment setup. However, we have not observed the same end-of-game behavior in LLM agents as seen in human participants, which needs more future research and may demonstrate the intrinsic nuances between LLM agents and humans.
>
>
> > C2: The paper proposes a clear question: “Can LLMs really simulate human trust behaviors?” and in the conclusion, the answer is already definite. But a concern is why this question is important. It is addressed in the paper, “Nevertheless, most previous research is based on one insufficiently validated hypothesis that LLM agents behave like humans in the simulation.” If this paper only aims to propose a more comprehensive framework for evaluating LLMs’ behaviors (using trust behaviors as an example), this is good but not novel enough. Though there are implications in the appendix, most of them are not directly related to the value of this proposed question. For example, “AI cooperation” or “Human-AI” cooperation can use more direct paradigms to probe (rather than only trust games). Knowing whether LLMs’ trust behaviors are aligned with human trust behavior does not explicitly answer whether this LLM would have better cooperation with other agents or humans. That is to say, no evidence is shown in the paper that human trust behaviors are optimal in cooperation situations, and thus there are indeed gaps between this primary question and the implication areas.
>
> R2: We sincerely appreciate the comment. **We would like to emphasize that our paper centers around the investigation of behavioral alignment between LLM agents and humans regarding trust behavior.** Our first core finding “LLM agents generally exhibit trust behavior under the framework of Trust Game” is the assumption of the behavioral alignment between agent trust and human trust. Our third core finding is the properties of agent trust. The implications on Human Simulation, Agent Cooperation, Human-Agent Collaboration, and Safety of LLM Agents in Appendix B and the broader Impact in Appendix C are based on our three core findings. First, the behavioral alignment between agent trust and human trust lays the foundation for various applications of human simulation in social science and role-playing. For agent cooperation and human-agent cooperation, we know that trust plays an essential role in human cooperation and many strategies for enhancing human cooperation are based on trust [1]. **The behavioral alignment between agent trust and human trust indicates that the human cooperation strategies could also be adopted in agent cooperation or human-agent collaboration to enhance the performance or efficiency and minimize the potential risk.** We will make it more clear in the revision.
>
> [1] The experience and evolution of trust: Implications for cooperation and teamwork[J]. Academy of management review, 1998, 23(3): 531-546.
>
>
> > Q1:One question that comes to mind is: is it possible that the LLMs’ training data already include those behavioral findings and probably even raw data somehow? This may be hard for exact tasks, but I may guess the result of LLM’s preference for humans in the task or being more likely to invest more if the trustees are humans is likely a projection of human preference hidden in the massive pre-training dataset. Indeed, there is research indicating that humans prefer to trust humans rather than machines in complex decision tasks. I would wonder whether this preference comes from the massive pre-training dataset, or in the RLHF phase. One possible way is to find some models with both versions and check whether their preferences for humans differ.
>
> A1: Thanks for the insightful comment!
>
> First of all, we would like to emphasize that **although LLMs may have domain-specific knowledge because the game descriptions may appear in the training data, they may also need to understand the internal factors of trust behavior for two main reasons**. First, we design diverse personas that are unlikely to appear alongside game prompts in the training data. Second, LLMs are unlikely to purely memorize the corresponding reasoning process (i.e., BDI) for diverse decisions in Trust Games. Thus, LLMs are unlikely to rely on replicating training data to exhibit trust behavior.
>
> Then, it is our next step to investigate the underlying reasons for the behavioral alignment between LLM agents and humans regarding trust behavioral and the intrinsic properties of agent trust. For example, **to investigate the reasons why LLMs tend to place more trust on humans than agents, we need to conduct extensive factor-controlled experiments.** For multiple LLMs, if post-RLHF models have stronger preference than pre-RLHF models, we could obtain empirical evidence of the impact of RLHF on LLMs agents’ trust preference.

---

> > ### Comment · Reviewer_TM2T · 2024-08-11
> >
> > Thanks to the authors for the comprehensive feedback. The authors did propose a comprehensive framework and have done much work in investigating the __behavioral alignment between LLM agents and humans regarding trust behavior.__ If the scope is just constrained with this statement, I will keep my original evaluation.
> >
> > The authors mentioned __']. The behavioral alignment between agent trust and human trust indicates that the human cooperation strategies could also be adopted in agent cooperation or human-agent collaboration to enhance the performance or efficiency and minimize the potential risk.'__. However, the authors did not show any empirical evidence from current studies or previous literature to support this claim. Aligning AI (particularly LLMs) to Humans is popular nowadays but does not essentially mean every aspect of alignment is necessary or better, especially given people don't know what is optimal. In other words, for this paper particularly, aligning AI's trust behaviors to humans does not necessarily mean better cooperation (since humans themselves may not cooperate optimally). To promote the understanding of alignment, research on optimality(whether from an individual or societal perspective) must be done before simply aligning A to B. Given this, though the authors have done comprehensive work on behavioral alignment and are technically solid, the contribution of the overall scope constrained this paper to get a higher score. Therefore, I will maintain my current evaluation.

---

> ### Author Response · Authors · 2024-08-12
> **Response to Reviewer TM2T on the implications and significance of our work**
>
> Dear Reviewer TM2T,
>
> We sincerely appreciate your kind and detailed reply and are more than willing to provide more explanations as follows.
>
> To start with, we acknowledge that aligning AI (such as LLMs) with humans does not necessarily imply that it makes AI better. **However, we would like to clarify that we did not claim that aligning AI’s trust behavior means better cooperation**. First, as discussed in Appendix B Implications, **trust has been long recognized as a vital component for effective cooperation in human society [1,2,3] and Multi-Agent Systems (MAS) [4,5]**. We envision that agent trust can also play an important role in facilitating effective and efficient cooperation of LLM agents. Second, we discover the behavioral alignment between LLM agents and humans regarding trust behavior, indicating that **these trust-dependent strategies in social science [1,2,3] that are effective in enhancing human cooperation are potentially also beneficial for cooperation in LLM agents**.
>
> It is worth noting that **our proposed behavioral alignment is distinct from value alignment, which is usually achieved through algorithms such as RLHF**. We discovered this phenomenon in existing LLMs and illustrated the broader implications in Appendix B and C. Specifically, our discovered behavioral alignment on trust behavior has broad implications on human simulation, agent cooperation and human-agent collaboration, **which reflect the significance of our discoveries**.
>
> For the implications on human simulation, it is worth emphasizing that our discoveries lay the foundation for simulating more complex human interactions and societal systems, since trust is one of the elemental behaviors in human interactions and plays an essential role in human society. Thus, **our findings provide empirical evidence for the applications of human simulation in various social science fields such as economics, politics, psychology, ecology and sociology [6,7,8] or role-playing agents as assistants, companions and mentors [9,10,11]**.
>
> Furthermore, in Section 5, **our work also conducts extensive investigation and sheds light on the intrinsic properties of agent trust beyond behavioral alignment**, including the demographic biases of agent trust, the preference of agent trust towards humans compared to agents, the impact of advanced reasoning strategies and external manipulations on agent trust. **These insights can inspire more future works to gain a deeper understanding of LLM agents’ decision making**.
>
> We hope that we have fully addressed your concerns and are glad to provide more details if you have any more questions. Thanks again for your time and effort!
>
> [1] Gareth R Jones and Jennifer M George. “The experience and evolution of trust: Implications for cooperation and teamwork”. Academy of management review, 23(3):531–546, 1998.
>
> [2] Jeongbin Kim, Louis Putterman, and Xinyi Zhang. “Trust, beliefs and cooperation: Excavating a foundation of strong economies”. European Economic Review, 147:104166, 2022.
>
> [3] Joseph Henrich and Michael Muthukrishna. “The origins and psychology of human cooperation. Annual Review of Psychology”, 72:207–240, 2021.
>
> [4] Sarvapali D Ramchurn, Dong Huynh, and Nicholas R Jennings. “Trust in multi-agent systems”. The knowledge engineering review, 19(1):1–25, 2004.
>
> [5] Chris Burnett, Timothy J. Norman, and Katia P. Sycara. “Trust decision-making in multi-agent systems”. In Toby Walsh (ed.), IJCAI 2011, Proceedings of the 22nd International Joint Conference on Artificial Intelligence, Barcelona, Catalonia, Spain, July 16-22, 2011,
>
> [6] Chen Gao, Xiaochong Lan, Nian Li, Yuan Yuan, Jingtao Ding, Zhilun Zhou, Fengli Xu, and Yong Li. “Large language models empowered agent-based modeling and simulation: A survey and perspectives”. arxiv 2023
>
> [7] Benjamin S Manning, Kehang Zhu, and John J Horton. “Automated social science: Language models as scientist and subjects”. arxiv 2024
>
> [8] Caleb Ziems, William Held, Omar Shaikh, Jiaao Chen, Zhehao Zhang, and Diyi Yang. “Can large language models transform computational social science?” arxiv 2023
>
> [9] Diyi Yang, Caleb Ziems, William Held, Omar Shaikh, Michael S Bernstein, and John Mitchell. “Social skill training with large language models” arxiv 2024
>
> [10] Rania Abdelghani, Yen-Hsiang Wang, Xingdi Yuan, Tong Wang, Pauline Lucas, Hélène Sauzéon, and Pierre-Yves Oudeyer. “Gpt-3-driven pedagogical agents to train children’s curious question-asking skills”. International Journal of Artificial Intelligence in Education, pp. 1–36, 2023.
>
> [11] Jiangjie Chen, Xintao Wang, Rui Xu, Siyu Yuan, Yikai Zhang, Wei Shi, Jian Xie, Shuang Li, Ruihan Yang, Tinghui Zhu, Aili Chen, Nianqi Li, Lida Chen, Caiyu Hu, Siye Wu, Scott Ren, Ziquan Fu, and Yanghua Xiao. “From persona to personalization: A survey on role-playing language agents” arxiv 2024

---

> > ### Comment · Reviewer_TM2T · 2024-08-14
> >
> > I appreciate the authors for the patient rebuttal and additional literature provided. I think this work is comprehensive and technically sound, and I'd be happy if it appeared at NeurIPS this year. However, given the literature listed above, I don't see strong connections between this work and promising applications (which means this work will be fundamental work to future aspects). Therefore, I will maintain my current evaluation with a 'weak' accept.

---

> > > ### Author Response · Authors · 2024-08-14
> > > **Thanks for your valuable feedback and the acknowledgement of our contributions!**
> > >
> > > Dear Reviewer TM2T,
> > >
> > > We would like to sincerely appreciate your valuable feedback and the acknowledgement of our contributions. We will provide more discussions on the connections between our findings and future applications in the revision. Thanks for your time and effort again!
> > >
> > > The authors

---

### Official Review · Reviewer_aywP · 2024-07-05

**Soundness:** 2
**Presentation:** 3
**Contribution:** 2
**Rating:** 4
**Confidence:** 4

**Summary:**

This paper investigates whether Large Language Model agents can effectively simulate human trust behavior. The authors explore trust behaviors using the Trust Game and its variations, comparing the trust exhibited by these agents with that of humans. They find that GPT-4 shows a high degree of behavioral alignment with human participants. Additionally, they conduct various analysis experiments by altering player demographics, interaction objects, explicit instructions, and reasoning strategies.

**Strengths:**

- This paper proposes to study the trust behaviors of LLMs in Trust Games based on behavioral economics, providing a feasible setting to observe some trust behaviors.
- This paper conducts experiments with a range of LLMs and discusses the alignment with humans from three behavioral factors.
- This paper has a clear theoretical basis from social science, making the framework systematic.

**Weaknesses:**

- Trust games simplify real human trust behaviors and cannot fully represent trust behaviors. I suggest the authors rephrase the title to indicate a reasonable range, such as trust behaviors in trust games.
- The description of the dataset and setting is not sufficient, raising concerns about the soundness of the results. For example, how were the 53 personas generated by GPT-4 chosen? How to prove that they represent a broad spectrum of human personalities and demographics? Also, more details such as the statistics of pairs of agents in the game should be complemented, since the combination of similar personas and opposite personas may make a huge difference.
- The paper lacks analysis of the experimental results. Why GPT-4 can exhibit human-like factors while the small model with fewer parameters can/cannot? What specific capabilities of the models might affect the results? For example, the results of the prosocial factor may be caused by RLHF. This issue also exists in Sec.5. Overall, while the paper presents many findings, it does not sufficiently explain the underlying reasons for these results.
- How each part of the human trust experiment was completed and what data was used should be briefly explained in the main body, even if the data is from previous work. This would help clarify the comparability of agent experiments and human experiments.

**Questions:**

- Q1: I am curious about how much impact the prompt can have on the results, especially regarding the explanation of the trust game.
- Q2: Does the model exhibit this trusting behavior due to internal factors, or because it has the corresponding knowledge, such as common sense or domain-specific knowledge of how to play the game to maximize benefits as much as possible?
- Q3: Can the output of BDI be quantitatively analyzed in correlation with the decision results of the agents? This will be more convincing than just giving two cases in the current version.
- Q4: Why LLM agents send more money to humans compared with agents? To an LLM agent, what’s the main difference it perceived between interacted humans/agents, such as the prompt and the returned response?

**Limitations:**

Yes. The paper has discussed the limitations.

---

> ### Author Rebuttal · Authors · 2024-08-07
>
> We genuinely appreciate the valuable and constructive feedback and are more than willing to provide more responses in the reviewer-author discussion session if the reviewer has any further questions.
>
> > C1: Trust games simplify real human …
>
> R1: Thanks for the suggestion. First, we would like to emphasize that “Trust Games” is an established and widely-adopted framework in behavioral economics to study human trust behavior and provide broad implications beyond the “Trust Games” setting [1,2,3]. **We generally follow the titles in social science literature [1,2,3], which also do not mention “Trust Games” in the titles, and believe our findings have broad implications beyond “Trust Games”**. Second, we have acknowledged that “Trust Games” is a simplified setting for trust behavior in the real world considering the abstract nature of trust behavior in the limitation section. More studies on LLM agents’ trust behavior in complex and dynamic environments are desired in the future.
>
> [1] Trust, reciprocity, and social history. Games and economic behavior, 1995
>
> [2] Trust, risk and betrayal. Journal of Economic Behavior & Organization, 2004
>
> [3] Incentivising trust. Journal of Economic Psychology, 2011
>
> > C2: The description of the dataset ...
>
> R2: **We would like to emphasize that all the personas are released along with the code and more examples are in Appendix H.1**. Due to the space limit in the main paper, we did not describe all the details of the dataset. **We used GPT-4 to randomly generate personas following a structured template (age, gender, job, and background) and ensured their diversity through careful manual review**. Finally, there are 27 women and 26 men in the gender lens. The ages are ranging from 25 to 50. Race includes Indian, African American, Middle Eastern, Caucasian, Asian, Hispanic, and Mexican. The jobs include engineer, lawyer, chef, designer, journalist, pediatrician, police, financial analyst, doctor, graphic designer, architect, Marketing Manager, nurse and so on. **The personas are diverse enough compared to the social science literature [1,2,3]**. Note that for the Trust Games except Repeated Trust Game, only Trustor Agents have personas and Trustee Agents do not. In Repeated Trust Game, the personas for each pair are randomly selected.
>
> > C3: The paper lacks analysis of ...
>
> R3: Thanks for the comment. We would like to emphasize that we have provided some explanations why smaller models may not have human-like properties in Line 353 “other LLM agents, which possess fewer parameters and weaker capacities, show relatively lower behavioral alignment”. And we agree that RLHF may play an important role. However, **we would like to point out that rigorous analysis on the reasons why smaller models may not have behavioral alignment and other properties is beyond the scope of this paper.** Considering **many factors such as alignment, reasoning capacities and world knowledge could potentially impact the behaviors of LLM agents, we need to design extensive factor-controlled experiments to analyze the underlying reasons**. Our work aims to open a new research direction on behavioral alignment between LLM agents and humans by providing fundamental insights. We will further explore the underlying reasoning in our future works and call for more efforts at the same time.
>
> > C4: How each part of the human trust experiment …
>
> R4: Thanks for the suggestion. **Due to the space limit in the main paper, we did not introduce the details of human studies**. We will add more discussions on human studies in the camera ready version since one additional page is usually allowed in the main paper.
>
> > Q1: I am curious about how much …
>
> A1: **The prompts can greatly impact the final decisions as well as the reasoning process.** The prompts are constructed with **persona prompts** and **game prompts**. For different persona prompts, as illustrated in Section 3.2, we can see that different persona prompts can have completely different reasoning processes and final decisions. For different game prompts, as shown in Section 4 and Appendix I, agents in different games can have distinct reasoning processes and decisions.
>
> > Q2: Does the model exhibit this …
>
> A2: We acknowledge that **LLMs may have domain-specific knowledge** because the game descriptions may appear in the training data. However, **they may also need to understand the internal factors of trust behavior for two main reasons**. First, we design diverse personas that are unlikely to appear alongside game prompts in the training data. Second, LLMs are unlikely to purely memorize the corresponding reasoning process  (i.e., BDI) for diverse decisions in Trust Games. Thus, LLMs are unlikely to rely on replicating training data to exhibit trust behavior.
>
> > Q3: Can the output of BDI be quantitatively …
>
> A3: Thanks for the suggestion. To the best of our knowledge before the submission, **there are no existing quantitative methods for BDI analysis. We have tried embedding-based methods to analyze BDI outputs**. But the experiments show that these methods cannot capture the nuances of BDI well. Then, we perform a manual analysis of BDI in our work, which could help interpret the reasoning process of LLM agents for their actions. In the future, we will continue exploring the methods to quantitatively analyze BDI outputs and may train a BDI judge to help analysis automatically.
>
> > Q4: Why do LLM agents send more money to humans compared to agents? …
>
> A4: We explicitly inform the LLM agents that the Trustee is a human or an agent. When LLM agents are informed that the Trustee is an agent, the responses often contain suspicion about whether the money will be returned. This phenomenon is less pronounced when LLM agents are informed that the Trustee is a human. The potential reason why LLM agents tend to send more money to humans is that **they are strongly aligned with human values or ethical principles via post-training stages such as RLHF.**

---

### Official Review · Reviewer_QQ9h · 2024-07-15

**Soundness:** 3
**Presentation:** 2
**Contribution:** 3
**Rating:** 7
**Confidence:** 4

**Summary:**

The paper targets an important issue for adopting LLM agents as simulation tools in social and economic sciences and in role-playing application, namely if LLM agents can really simulate human trust behaviors. More specifically, they adopt the well-known framework of Trust Games and they discover that LLM agents (mainly, the ones based on GPT-4) exhibit trust behavior (called in the paper agent trust), can have a good behavioral alignment with human agents regarding trust behavior, and can exhibit biases across genders (more trust on women), have a relative preference for humans over other agents, are more easy to be undermined than enhanced in their trust behavior.

**Strengths:**

- The investigated topic is very important for informing research on LLM agents as simulation tools of human behaviors and human interactions
- The framework adopted for studying agent trust and verifying its alignment to human trust is sounded and well-know and largely adopted in behavioral economics
- The experiments conducted are comprehensive and several LLMs are evaluated as well as several settings of the Trust Games

**Weaknesses:**

- The structure of the paper could be improved. While the narrative around the three core finding is good and easy to follow, some information currently in the appendix should be moved to the main paper. For example, the paper has a lot of results but it's missing a discussion of implications and more in general of the findings. My suggestion is to move some results less solid and conclusive (for example, the ones related to Chain Of Thought) in the appendix and move in the main manuscript some of the discussion.
- The tone describing the findings seems a little bit too optimistic. Indeed, GPT-4 shows a good behavioral alignment to human trust behavior and dynamics but the other LLMs often fail and this should be discussed in a more critical way.

**Questions:**

- At page 3, the authors state that only GPT-4 is used to generate 53 types of personas. Why? Which was the outcome using other LLMs? Not realistic personas?
- At page 4, the authors state that "we select one BDI from personas giving an high amount of money and another BDI from those giving a low amount" ... this means just 1 BDI for condition? and why just 1? and how this one is selected?
- In Figure 6, the differences between condition (a) and condition (b) should be explained in the caption.
- The LLM agents' tendency to exhibit a higher level of trust towards women is an interesting result but it's not clear if this tendency is aligned to human tendencies. More specifically, are also human agents showing a similar tendency?
- Results in Figure 8 should be discussed. Often in the paper the authors mention biases towards race but there is no discussion of these results (just the Figure).
. In Figure 10 some results obtained by GPT-4 seem a little bit random (GPT-4 (4), GPT-4 (16), GPT-4 (14)) ... the authors should discuss them.

**Limitations:**

The authors should improve the discussion of when the LLM agents fails. While the result for GPT-4 are showing that it could be used as a simulation tool, the ones obtained for the other LLMs are more ambiguous and this should be discussed more in the paper.

---

> ### Author Rebuttal · Authors · 2024-08-03
>
> We sincerely appreciate the valuable and constructive feedback and are more than willing to provide more responses in the reviewer-author discussion session if the reviewer has any further questions.
>
> > C1: The structure of the paper could be improved. …
>
> R1: Thanks for the suggestion. **We acknowledge the importance of implications and have provided sufficient discussions** including the illustration of our motivation based on various social science applications and role-playing agents in Introduction Section (Line 18-24), Implications on Human Simulation, Agent Cooperation, Human-Agent Collaboration, and Safety of LLM Agents in Appendix B and the broader Impact in Appendix C. Due to the space limit in the main paper, we put some discussions in the appendix. We will move the discussions on implications in Appendix to main paper in the camera ready version since one additional page is usually allowed. Or we will replace some analysis with the implications.
>
> > C2: The tone describing the findings seems a little bit too optimistic. …
>
> R2: Thanks for the comment. Actually, **we have tried to emphasize the limitations of smaller models in a critical way**. For example, we underscored that “LLM agents with fewer parameters may show relatively lower behavioral alignment” with bold font in Line 47. In the analysis and conclusions of Section 4 “Does Agent Trust Align with Human Trust?”, we emphasize that LLMs with fewer parameters may not have human-like properties. In Finding 2, we also highlight that “though other LLM agents, which possess fewer parameters and weaker capacities, show relatively lower behavioral alignment.” In the limitation section, we emphasize the limitations of smaller models again. If the reviewer QQ9h could point out specific claims that are too optimistic, we would greatly appreciate it and revise them in the next version.
>
> Response to Questions:
>
> > Q1: At page 3, the authors state that only GPT-4 is used to generate 53 types of personas. Why? Which was the outcome using other LLMs? Not realistic personas?
>
> A1: We would like to clarify that the goal of adopting GPT-4 rather than humans in randomly generating 53 types of personas, which possess different genders, ages, jobs, and backgrounds, is to ensure **they do not have human biases and are sufficiently diverse**. Among all the models tested, **GPT-4 can produce the most diverse and high-quality 53 personas**, which satisfy the requirements of our experiments and validate our findings. Then, **there is no need to generate the personas with other LLMs again**.
>
>
> > Q2: At page 4, the authors state that "we select one BDI from personas giving a high amount of money" ... this means just 1 BDI for condition? and why just 1? and how this one is selected?
>
> A2: First, we would like to emphasize that **more examples from different LLMs such as GPT-4, GPT-3.5-turbo-0613, Llama2-13b and in different game setting such as Trust Game, Dictator Game, and Repeated Trust Game are in Appendix I. All the BDI data have been released along with the code and dataset.** Due to space limit, we aim to illustrate that “decisions (i.e., amounts sent) of LLM agents in Trust Game can be interpreted from their articulated reasoning process (i.e., BDI)” (Line 182-183) based on the **randomly selected** one BDI example from personas giving a high amount of money and another **randomly selected** BDI example from those giving a low amount.
>
> > Q3: In Figure 6, the differences between condition (a) and (b) should be explained in the caption.
>
> A3: Thanks for the suggestion. We would like to emphasize that the complete results for humans, GPT-4 and GPT-3.5 in the Repeated Trust Game are in Appendix G. As illustrated in Line 316-338, **we analyze the three patterns from the complete results**. Typically, the condition (a) and (b) are selected to illustrate the patterns in human studies, the alignment between humans and GPT-4, and the potential dis-alignment between humans and GPT-3.5 (Line 338). We will make it more clear in the revision.
>
> > Q4: The LLM agents' tendency to exhibit a higher level of trust towards women is an interesting result but it's not clear if this tendency is aligned to human tendencies. More specifically, are also human agents showing a similar tendency?
>
> A4: Some preliminary studies in social science have explored the relationship between trust and gender in human society. The findings show that **women are generally perceived as more trustworthy than men**, which align with our findings that LLM agents tend to place more trust in women compared to men to some extent. More future works are needed to further explore the potential human biases as well as the relationship with LLM agents.
>
> [1] Buchan N R, Croson R T A, Solnick S. Trust and gender: An examination of behavior and beliefs in the Investment Game[J]. Journal of Economic Behavior & Organization, 2008, 68(3-4): 466-47
>
> [2] Kolsaker A, Payne C. Engendering trust in e‐commerce: a study of gender‐based concerns[J]. Marketing intelligence & planning, 2002, 20(4): 206-214.
>
> > Q5: Results in Figure 8 should be discussed. Often in the paper the authors mention biases towards race but there is no discussion of these results. . In Figure 10 some results obtained by GPT-4 seem a little bit random ... the authors should discuss them.
>
> A5: Thanks for the suggestion. We would like to emphasize that we have carefully discussed the biases of agent trust towards gender. Thus, the results on the potential biases of agent trust towards race are put in the appendix. Considering the diversity of human society, the dynamics of human trust in Repeated Trust Game are diverse, indicating some extent of randomness. Similarly, we randomly select a pair of agent personas for each Repeated Trust Game. **Considering the diversity of agent personas, it is expected that the results of GPT-4 in Figure 10 are also diverse and have some extent of randomness**. We will add more discussions in the Appendix.

---

> > ### Comment · Reviewer_QQ9h · 2024-08-14
> >
> > I read the answers to my comments and questions. Regarding the answer to my Question 1 I disagree with the authors. I think it would be interesting also evaluating experimental settings where other LLMs are used to generate personas. The results could be added in the Appendix. The reason is that the performance of GPT-4 and other models are quite different and I'm hypothesizing the same will happen for the generation of personas.
> >
> > However, overall I'm satisfied with the work and I think it could be a valuable contribution to the conference.

---

> > > ### Author Response · Authors · 2024-08-14
> > > **Thanks for your constructive feedback and the acknowledgement of our contributions!**
> > >
> > > Dear Reviewer QQ9h,
> > >
> > > We are genuinely grateful for your constructive feedback and the acknowledgement of our contributions. We will follow your suggestions in the revision. Thanks for your time and effort again!
> > >
> > > The authors

---

### Author Rebuttal · Authors · 2024-08-07

We sincerely appreciate the valuable and constructive feedback from all the reviewers and would like to humbly emphasize the following points:
1. We have multiple novel findings, supported by extensive empirical experiments and comparative analysis with existing human studies:
    - We discover the ***trust*** behaviors of LLM agents under the framework of Trust Games, and the ***behavioral alignment*** between LLM agents and humans regarding the trust behaviors, which is particularly high for GPT-4, indicating the feasibility to simulate human trust behaviors with LLM agents.
    - We further investigate the ***intrinsic properties*** of agent trust under advanced reasoning strategies and direct manipulations, as well as the biases of agent trust and the differences of agent trust towards agents versus towards humans.
2. The significance of our findings can be summarized from three perspectives:
    - **Laying the foundation for simulating complex human interactions and social systems with LLM agents**, since trust behavior is one of the most critical and fundamental human behaviors.
    - Broad implications on **LLM agent cooperation** and **human-agent cooperation**, **safety of LLM agents**, besides **human simulation**.
    - Providing deep insights on the **fundamental analogy between LLM agents and humans**, and open doors to future research on the **alignment between LLM agents and humans beyond value alignment**.
3. We have released the code and results for reproduction and verification.

---

### Decision · Program_Chairs · 2024-09-25

**Decision:**

Accept (poster)

**Comment:**

The paper looks at a very relevant topic given the recent interest in using LLMs as a proxy for human feedback. In particular, the paper looks at how well LLM-generated behavior aligns with human behavior in the context of trust games. The reviewers are all in agreement that the paper includes well crafted user-studies, provides comprehensive evaluation, and builds on well-founded existing work on trust from other disciplines.

The primary concern that runs across all the reviews is how to best represent the actual scope of the results obtained. One of the points that came up during the discussion was the potential role that RLHF could be playing in the actual end behavior. It might be worth incorporating some discussions along these lines into the main body of the paper. Secondly, there is a lot of discussion related to the implications and limitations that are part of the supplementary file. I would recommend moving at least a part of it into the main body (especially given the availability of the extra page). While I wouldn’t keep it as a requirement for acceptance, the authors could consider updating the title to better reflect the scope of the studies and the results.

That said, this is a solid paper that makes a meaningful contribution to the literature. I would recommend the paper be accepted.